# Host response biomarkers of tuberculosis recurrence and treatment failure
Bernadette Bauer [1] ✉, Mohamed I. M. Ahmed[1,2], Olga Baranov[1,2,3], Abhishek Bakuli[1,2], Luming Lin[1,2], Abisai Kisinda[4], Mkunde Chachage[1,4,5], Nyanda E. Ntinginya[4], Celso Khosa[6], Michael Hoelscher [1,2,3,7], Mohammed Rassool[8], Salome Charalambous[9,10], Jayne S. Sutherland[11], Kathrin Held [1,2,3,7,18], Andrea Rachow[1,2,7,18] & Christof Geldmacher[1,2,3,18] On behalf of the TB sequel consortium*

## Abstract

**Background** Accurate detection of tuberculosis (TB) treatment failure and recurrence can improve disease control, but current sputum-based monitoring tools pose significant limitations. This study aimed to identify sputum-independent biomarkers for detecting and predicting TB treatment failure and recurrence.

**Methods** Within the Pan-African TB Sequel study, we conducted a matched case-control study with 40 participants who had recurrent TB or treatment failure and 37 successfully treated controls matched by sex, age, and HIV status. Cases were classified as (a) non-converters with persistently positive sputum *Mycobacterium tuberculosis* (MTB) results during treatment, (b) reverters at the end of treatment (EOT), or (c) recurrence after EOT. Peripheral blood was collected at baseline, months 2, 4, 6, 9, and 12, and at suspected recurrence. MTB-specific T-cell activation markers (CD38, CD27, HLA-DR, Ki67) and transcriptomic signatures (Sweeney3, Risk6, MAMS6) were assessed and compared to the reference standard MTB culture and smear results.

**Results** Here, we show that both MTB-specific T-cell activation and transcriptomic signatures detected non-conversion and TB recurrence at month 9 or 12 after treatment initiation. CD38 expression demonstrates 100% sensitive (95% CI: 56.6–100%) and 78% specific (95% CI: 56.5–99.4%) for detecting TB recurrence, with an AUC of 0.98 (95% CI: 91–100%). Among transcriptomic signatures, MAMS6, RISK6, and Sweeney3 achieve 75% sensitivity (95% CI: 50–100%) and 87–93% specificity (95% CI: MAMS6 0–100%, RISK6 0–93%, Sweeney3 0–100%), with comparable AUCs (0.78–0.83). Neither marker detected TB reversion at EOT.

**Conclusion** These sputum-independent biomarkers effectively identify TB disease, non-conversion and recurrence TB after EOT, whereas their utility in detecting TB reversion during treatment remains limited.

## Plain Language Summary

Tuberculosis (TB) is a serious infectious disease that can be fatal if untreated. While most patients recover with treatment, some do not respond well or develop TB again after completing therapy. Monitoring how well patients respond to TB treatment currently relies on tests using sputum samples, which can be slow and may be less reliable during treatment. This study aimed to identify alternative host-based markers in blood that could help detect patients with poor treatment response or TB recurrence. We found that specific blood markers can reliably identify patients with TB and detect poor treatment response during therapy, as well as TB recurrence after treatment completion. These findings may help improve early detection, guide treatment decisions, and reduce TB transmission.

Tuberculosis (TB) disease, caused by the acid-fast bacillus *Mycobacterium tuberculosis* (MTB), continues to pose a significant global health challenge. In 2023, an estimated 10.8 million individuals developed TB and 1.25 million deaths attributable to TB were reported worldwide[1]. Despite the availability of treatment, the global success rate for first-line regimens remains suboptimal at 88%[1], for a variety of reasons such as the lengthy duration of therapy, side effects and poor tolerability[2]. Treatment failure

occurs in 0.72% of patients with drug-susceptible TB globally[1], while TB recurrence after successful treatment completion has an estimated incidence rate of 2.26 per 100 person-years[3], further complicating efforts to achieve lasting disease control. Identification of TB patients with a high likelihood of treatment failure and disease recurrence will hence help reduce drug resistance by allowing for treatment adjustments stopping ongoing bacillary replication under suboptimal therapy that might select for resistant TB

A full list of affiliations appears at the end of the paper. *A list of authors and their affiliations appears at the end of the paper. ✉e-mail: bauer.be@campus.lmu.de; bauer.bernadette@web.de

strains[2,4,5]. Further, it will aid in decreasing transmission, minimizing post-treatment adverse events, and lowering healthcare costs.

Established risk factors, such as a positive 2-month sputum culture or smear, are important for predicting tuberculosis treatment outcomes, but do not reliably differentiate between individuals with unfavorable treatment outcomes and those who achieve a cure[6–10]. Moreover, sputum-based diagnostics have limitations, particularly in individuals with extrapulmonary TB, those unable to produce sputum (e.g., children), and patients with paucibacillary disease. Sputum-based MTB diagnostics, in particular MTB culture, also requires a higher biosafety level[11]. Hence, there is a compelling need to advance diagnostic tools for early sputum-independent diagnosis of TB treatment failure and recurrence.

Several blood biomarkers have shown promising results to diagnose TB and are also gaining recognition for treatment monitoring[12–18]. The activation and maturation status of MTB-specific CD4 T cells distinguishes quite accurately between individuals with TB disease, clinically "latent" MTB infection (TBI), and cured TB[12,13,19,20]. Furthermore, after treatment initiation, the decline in MTB-specific T-cell activation correlates with time to culture conversion. Moreover, such activated T cells can often be detected even before symptom onset and before MTB bacilli become detectable in sputum[19,21].

Transcriptomic profiling has emerged as another powerful tool not only for diagnosing TB disease[22–29] but also for predicting progression to TB disease[14,16,22,30,31] and monitoring treatment outcomes[17,18,32–36]. Current host RNA signatures are often composed of only a few transcripts, facilitating their translation to RT-qPCR (Reverse Transcription quantitative Polymerase Chain Reaction), making them suitable for point-of-care platforms. However, due to the limited occurrence of TB treatment failure and recurrence, these RNA signatures have yet to be extensively tested across diverse cohorts for their ability to predict or diagnose TB recurrence.

To address whether and when sputum-independent biomarkers can predict and detect treatment failure and recurrence TB, we conducted a matched case-control study nested within the TB Sequel study, a large multicenter cohort in West, East and South Africa. The cohort follows TB patients intensively for 24 months after microbiological diagnosis[37], with extended follow-up every 6 months thereafter for up to 5 years. We studied the dynamics of MTB-specific T-cell activation markers in PBMC using flow cytometry with pre-defined cut offs, and three parsimonious transcriptomic signatures in whole blood from patients with TB treatment failure or recurrence and compared these with successfully treated patients over time, who were matched by sex, age and HIV status. Close monitoring of microbiological outcomes enabled the identification of distinct subgroups: non-converters, reverters at EOT, and recurrence after EOT, and allowed for a detailed assessment of the diagnostic and predictive potential of biomarkers in each subgroup.

We show that specific T-cell activation markers and transcriptomic signatures reliably identify patients with TB disease, microbiological non-conversion and recurrence TB after EOT. These findings highlight the potential of blood-based biomarkers to detect TB recurrence after treatment completion, providing a sputum-independent tool for improved patient monitoring and risk stratification.

## Methods

### Study participants and sample selection

Patients with microbiologically confirmed pulmonary tuberculosis based on MTB positive Xpert MTB/RIF assay or positive sputum culture were recruited as part of the longitudinal TB Sequel study[37]. This prospective, multi-country (South Africa, Tanzania, Mozambique, The Gambia), multi-center, observational cohort study recruited 1.430 pulmonary TB patients from 29 August 2017 until 27 December 2019. After enrollment, participants received standard tuberculosis treatment according to local TB treatment guidelines. The TB Sequel study followed TB patients through multiple visits: screening (days −6 to −1); baseline (BL), defined as the day of treatment initiation or up to 7 days afterward; day 14; months (M) 2, 4, and 6; and post-treatment visits at M9, M12, M18 and M24 after enrollment, with additional follow-up every 6 months for up to 5 years. Sputum samples for microbiological testing were routinely collected at screening, BL, M2, M4, and M6. After the M6 follow-up, sputum testing was only performed if TB recurrence was clinically suspected based on signs and symptoms. Peripheral blood mononuclear cells (PBMCs) and PAXgene blood samples were collected at BL, M2, M4, M6, M9, M12, and in cases of suspected recurrence.

Clinical assessments in the study included chest X-rays, physical examination, TB-specific symptoms questionnaires, treatment response and adherence questionnaires. A detailed description of the study methodology is available in Rachow et al.[37]. All study participants provided written informed consent. The study protocol was approved by the ethics committees of the following institutions: Human Research Ethics Committee of the University of the Witwatersrand, Johannesburg, South Africa (Ref. No. 161111); Comité Nacional de Bioética para a Saúde, Maputo, Mozambique (Ref. No. 200/CNBS/22); The Gambia Government/MRC Joint Ethics Committee, The Gambia (Ref. No. 26487); Mbeya Medical Research and Ethics Committee, Mbeya, Tanzania (Ref. No. SZEC-2439/R.C/V.1/37); and the Ludwig-Maximilians-Universität München Ethics Committee, Munich, Germany (Ref. No. 786-16)[37].

In an interim analysis at the end of March 2022, 50 drug-susceptible TB patients were diagnosed with recurrent tuberculosis. Cases of treatment failure and recurrence were identified either during the 6-month treatment period or after treatment completion (end of treatment, EOT) during the follow-up phase. This analysis included all recurrence patients identified in the interim analysis with available blood samples ($n = 40$): 20 participants provided peripheral blood mononuclear cells (PBMC) for T cell activation marker (TAM-TB) analysis, and 37 participants provided peripheral blood samples using PAXgene for transcriptomic analysis. 30 and 13 participants had no available PBMCs and PAXgene tubes for TAM-TB analyses or transcriptomic analysis, respectively. Unfortunately, all Gambian samples for TAM-TB analysis ($n = 12$ recurrence cases and $n = 12$ controls) were lost due to a dysfunctional dry shipper during international shipping, while the clinical site in Johannesburg did not collect PBMC samples during this study. The control group ($n = 38$) (TAM-TB: $n = 21$, RNAseq: $n = 36$) completed TB treatment with no detectable MTB in culture or smear microscopy and showed no clinical signs of tuberculosis during a mean follow-up period of 35 months (range: 12–48 months) after treatment initiation. Due to poor sample quality, two participants from TAM-TB and one from RNAseq analysis were excluded, leaving 37 participants (TAM-TB: $n = 19$, RNA-seq: $n = 35$) for analysis.

All individuals included in our substudy had drug susceptible tuberculosis at baseline. TB treatment failure and recurrence cases were re-treated for TB either based on microbiological confirmation ($n = 34$) or clinical diagnosis ($n = 6$). We divided the treatment failure and recurrence TB group into three subgroups based on the timing of recurrence[1]: Non-converters, patients who continuously had a positive sputum smear or culture result during the entire 6 months of treatment[2]; Reverters at EOT, patients with positive microbiologically findings at M6 but conversion to negative microbiological results in smear and culture during treatment[38]. Notably, the generic term of "treatment failure" is used for both aforementioned groups. The third subgroup, recurrence after EOT, had negative microbiological results at EOT but subsequently developed signs of TB recurrence with positive microbiological findings, which may represent either relapse or reinfection (Fig. 1).

### Bacteriological assessment

Decontaminated and concentrated sputum samples were analyzed by smear microscopy using Ziehl-Neelsen staining. AFB-positive sputum samples were graded according to the WHO/IUATLD scale. Moreover, sputum samples were cultured on solid Lowenstein-Jensen (LJ) media (solid culture, SC) and in liquid media using the BACTEC MGIT 960 system (liquid

**Fig. 1 | Flowchart of the selection process for TB treatment failure and recurrence cases and successfully treated controls, matched by site, sex, age, and HIV status.** The treatment failure and recurrence group include: 1. non-converters (positive culture/smear results throughout treatment), 2. reverters at EOT (positive culture/smear at months 6 despite prior negative results). Subgroups one and two are summarized as "TB treatment failure". 3. Recurrence after EOT (recurrence (positive culture/smear) after treatment completion and negative culture/smear at EOT). The control group consisted of tuberculosis patients who were successfully treated and remained disease-free throughout the follow-up period. Note: Exclusions were due to missing or poor-quality samples. Many missing samples in the TAM-TB analysis were caused by shipping issues and lack of PBMCs in South Africa. EOT end of treatment, TB tuberculosis, PBMC peripheral blood mononuclear cells, TAM-TB T-cell activation marker TB assay, RNAseq RNA sequencing.

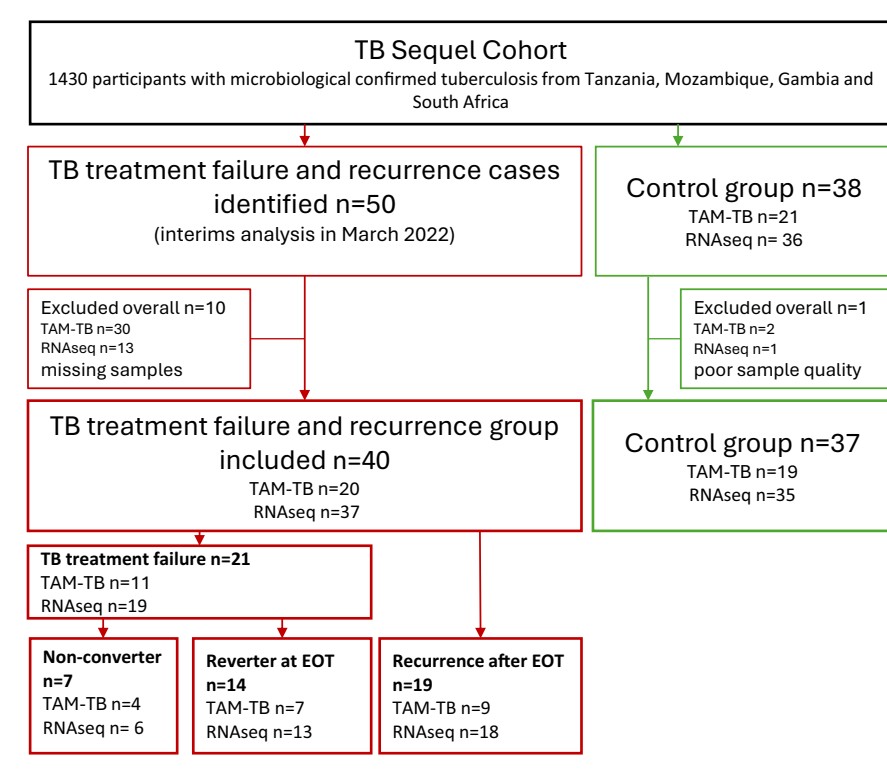

culture, LC). Positive cultures were confirmed for *Mycobacterium tuberculosis* (MTB) using the TB Ag MPT64 Rapid Test[37].

## Selection of PBMC samples and intracellular cytokine staining

Cryopreserved peripheral blood mononuclear cell (PBMC) samples from treatment failure, recurrence TB patients and matched controls were collected at the Tanzanian and Mozambican study sites. Intracellular cytokine staining of PBMCs was performed to determine MTB-specific CD4 T-cell activation and maturation using a protocol adapted from[19]. Briefly, PBMCs were stimulated overnight using Purified protein derivative (PPD, 10 μg/ml, Serum Staten Institute), MTB125 (2 μg/ml/peptide, peptides&elephants, Germany), staphylococcal enterotoxin B (SEB, 0.6 μg/ml, Sigma-Aldrich) as positive control, or no added peptides as negative control. Of note, MTB125 is a peptide pool consisting of 125 selected immunogenic MTB peptides described in more detail by Lindestam Arlehamn et al.[39]. Additionally, co-stimulatory antibodies anti-CD28 (L293, BD; 0.25 μl per reaction, 1 μg/ml final concentration, stock 1 mg/ml) and anti-CD49 (L25, BD; 0.25 μl per reaction, 1 μg/ml final reaction, stock 1 mg/ml) as well as Brefeldin A (BFA, final concentration 5 μg/ml, Sigma) were added to each sample. Cell surface staining was performed for 20 min with anti-CD38-BV785 (clone HIT2, Biolegend; 1.25 μl per reaction), anti-CD4-APC (clone 13B8.2, Beckmann Coulter; 5 μl per reaction), anti-CD27 ECD (clone 1A4CD27, Beckmann Coulter; 2.5 μl per reaction), and anti-HLA-DR APC-H7 (clone G46-6, Becton Dickinson; 1.25 μl per reaction), followed by fixation and permeabilization using FoxP3 Perm/Fix buffer and diluent (eBioscience), and then stained intracellularly using anti-IFNγ FITC (clone B27, BD Pharmingen; 1.0 μl per reaction), anti-Ki67 BV421 (clone B56, BD Pharmingen; 1.25 μl per reaction), and anti-CD3 APC-A700 (clone UCHT1, Beckmann Coulter; 2.5 μl per reaction). Cells were acquired on a CytoFlex flow cytometer (Beckman Coulter). Data analysis was done using the CytExpert software (Beckman Coulter). A positive MTB-specific CD4 T cell response was defined by an IFNγ response to the positive control antigen, a frequency of MTB-specific IFNγ + CD4 T cells ≥0.02%, and a count of ≥15 IFNγ + CD4 T cells after MTB125 or PPD stimulation, accounting

for background subtraction. Gating analysis was conducted blind to TB diagnosis (Supplementary Fig. 1).

## Determination of TAM-TB cut off to define active TB disease

In this study, the expression of CD38 on MTB-specific CD4 T cells was used to monitor treatment response over time and differentiate between treatment failure or recurrence TB and cured TB. The cutoffs for the frequency of CD38+ among total IFNγ + MTB-specific T cells used to identify TB disease were established through ROC analysis based on data from two previously published African studies[19,21], which applied the same ICS analysis and PPD restimulation methods[19,21]. We found that setting the threshold at 31.6% CD38+ of total IFNγ + MTB-specific T cells discriminated best between TB disease and latent TB infection with a sensitivity of 87% and specificity of 90% (Supplementary Fig. 2). This cutoff was applied to define TB disease in the current study, as illustrated in the color code of Fig. 3, in Figs. 2, 4, and Supplementary Fig. 4.

## RNA processing, sequencing, and signature calculation

Whole blood was collected from TB Sequel participants into PAXgene blood RNA tubes (BD Biosciences) and subsequently stored at −80 °C. RNA was extracted from PAXgene tubes using PAXgene Blood miRNA kits (Quiagen). RNA concentration was measured using the Xpose Spectrophotometer (Trinean). cDNA libraries were generated using NEBNext Ultra II Directional RNA kits (New England Biolabs) following the kit's instructions. After a final QC, the libraries were sequenced in a paired-end mode (2 × 100 bases) in the Novaseq6000 sequencer (Illumina) with a depth of ≥50 million reads per sample.

Underlying data derived from paired-end RNA sequencing of whole blood of the patients. The reads were mapped using kallisto package (version 0.48.0)[40] to the reference genome version GRCh38.p12 followed by subsequent in silico globin depletion. Genes containing less than 10 reads overall were excluded. Final set of patients for whom the sequencing data was available consisted of 35 cured and 36 patients experiencing recurrent TB. Genes were matched to the signatures using their ENSEMBL ID. As no established method for conversion of RNA-seq counts to CT values exists, rlog normalized counts were used (rlog method from DESeq2 package version 1.36.0)[41]. Tested and shown signatures are: Sweeney 3[24], RISK6[16] and

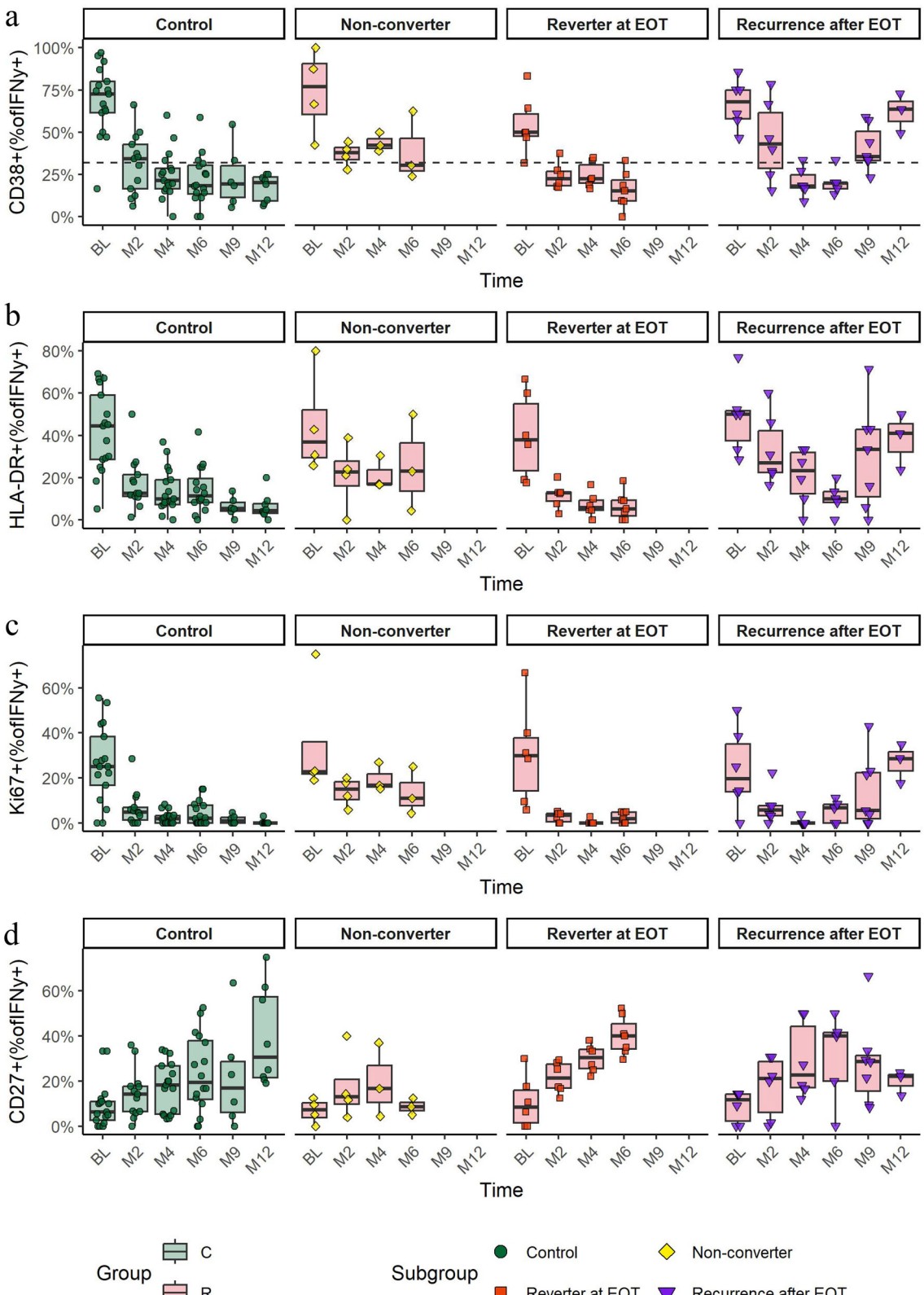

**Fig. 2 | Longitudinal phenotypic profiles of MTB-specific CD4 T cells in recurrence subgroups and control groups.** Activation marker expression is shown for CD38 (**a**), HLA-DR (**b**), Ki67 (**c**), and the maturation marker CD27 (**d**) across six time points (x-axis) from treatment initiation (BL) to 6 months post-treatment (M12), excluding any samples collected after TB recurrence. Color and shape codes indicate subgroup classification: treatment failure and recurrence group (non-converter, reverter at EOT, recurrence after EOT) and the control group. Sample sizes were as follows: non-converter $n = 4$, reverter at EOT $n = 7$, recurrence after EOT $n = 9$, control group $n = 19$; note that not all participants had samples available at every time point. MTB-specific CD4 T cells were characterized after stimulation with MTB 125. The box plot displays the interquartile range (IQR), with whiskers extending to the furthest points within 1.5 times the IQR. The dashed line in A indicates the cutoff for active tuberculosis, analyzed only for CD38. EOT End of treatment, BL Baseline, M Month, C Control group, R Recurrence and treatment failure group (Non-converter, Reverter at EOT, Recurrence after EOT).

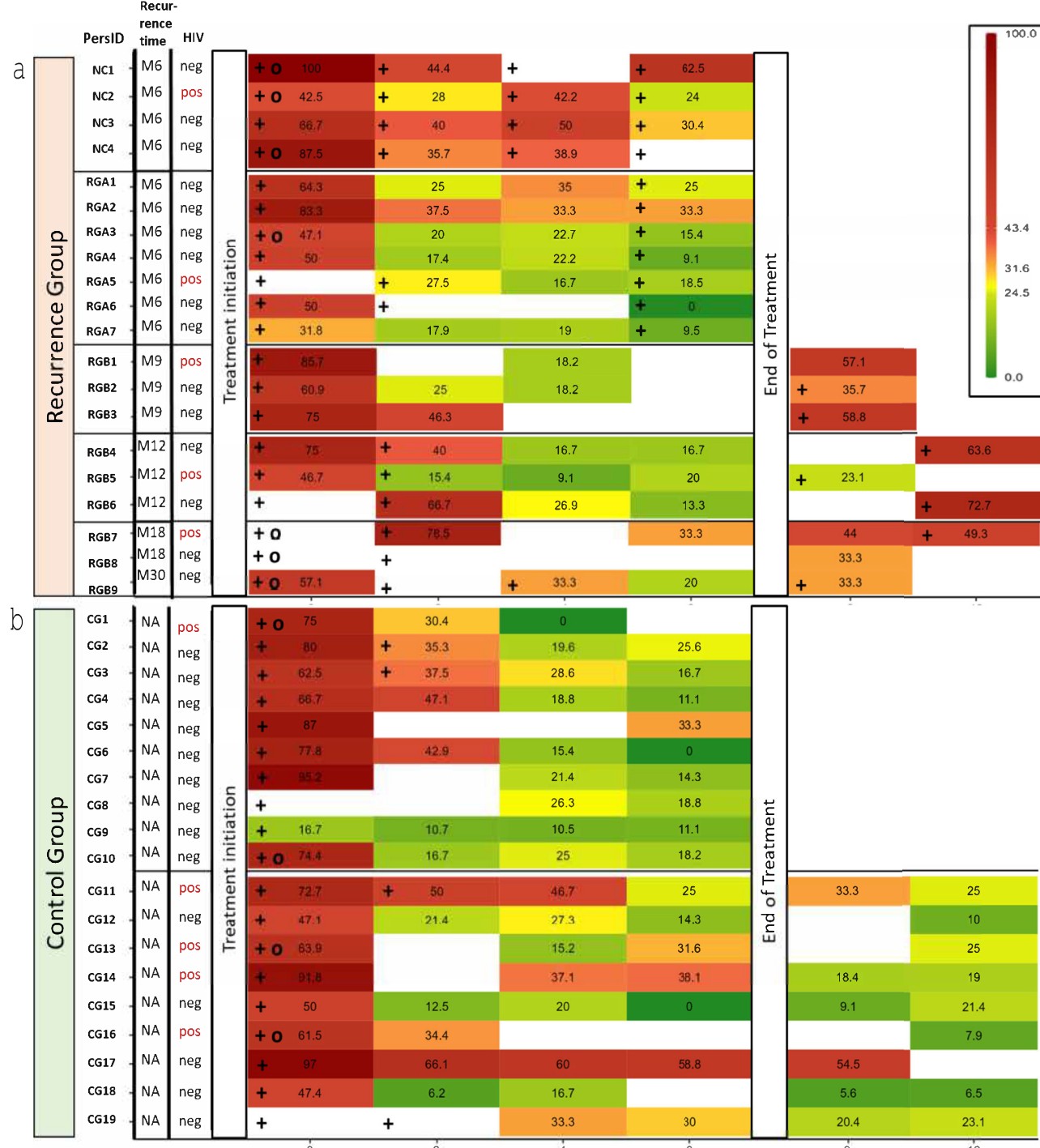

**Fig. 3 | Longitudinal frequencies and clinical characteristics of CD38+ MTB-specific IFNγ+ CD4 T cells during and after tuberculosis treatment.** Frequencies of CD38+ MTB-specific IFNγ+ CD4 T cells in patients with unfavorable treatment outcome (**a**, recurrence group) and successfully treated patients (**b**, control group) are shown from treatment initiation up to 6 months post-treatment (x-axis, study months). Positive microbiological results are marked with a "+" symbol. Black circles indicate cavitary disease at baseline. HIV status is indicated for all participants, and "Recurrence time" shows the month of recurrence/reverter diagnosis (not applicable for controls, indicated by "NA"). The heat map colors represent the percentage of activated MTB-specific T cells, with red indicating higher activation and green lower activation. PersID Participant Identification Number, NC non-converter, RGA Reverter group A (reverters at EOT), RGB Recurrence group B (recurrence after EOT), CG Control group, HIV Human Immunodeficiency Virus, neg negative, pos positive.

MAMS_6 (Ahmed et al. 2025, under review). The calculation was performed using the formulas and gene IDs in Supplementary Table 1; gene names in the formula represent log normalized counts. No significance testing was performed due to low numbers of subgroups. Performance of the signature was determined using Youden's J statistic. AUCs were calculated using pROC package (1.18.5)[42]. Published cut-offs from the three signatures were established using qPCR derived CT values, hence these cut offs were not applicable to RNA sequencing data. Nonetheless, we expect that the signatures will show a degree of separation between the groups when calculated from RNA sequencing and microarray data comparable to qPCR. Additional packages used for data handling were tidyverse (v. 2.0.0) and vroom (v. 1.6.5) packages.

## Statistical analysis

In Table 1 and Supplementary Table 1, continuous variables are summarized using the median and interquartile range (IQR). Statistical comparisons for continuous variables were performed using the Wilcoxon rank-sum test or the Wilcoxon signed-rank test, as appropriate. Categorical variables are presented as counts (n) and percentages, with statistical comparisons conducted using either Fisher's exact test or Pearson's chi-squared test, depending on the data. The Wilcoxon rank-sum test was used to compare the phenotypic profiles of MTB-specific CD4 + T cells between recurrence cases and controls (Fig. 2, Fig. 4, Supplementary Fig. 4). For comparisons of frequency and phenotypic profiles of MTB-specific CD4 + T cells stimulated with MTB125 or PPD across all participants, the Wilcoxon signed-rank test was applied (Supplementary Fig. 3). RNA analysis with DESeq included Wald testing and FDR correction and was applied as implemented in the DESeq2 package.

Statistical analyses, were performed in R[43], and figures were created using ggplot2[44]. Besides GraphPadPrism 8.4.2. was used for Receiver Operating Characteristic (ROC) analysis in Supplementary Fig. 2. A p-value ≤ 0.05 was considered significant. Relevant details on tests used are provided in the figure legends.

## Results

### Demographic and clinical characteristics of study participants

By the end of March 2022, 50 drug-susceptible TB patients were diagnosed with recurrent tuberculosis based on the initiation of retreatment. Cases were classified as treatment failure if occurring during treatment, or as recurrence if identified after EOT. In total, our analysis included 40 treatment failure and recurrent TB cases with a median recurrence time of 9 months (IQR 9–12). The TAM-TB dataset comprised 20 treatment failure or recurrence cases and 19 controls from Tanzania and Mozambique, while the transcriptomic dataset included 37 treatment failure or recurrence cases and 35 controls from all four countries (Fig. 1). The TB treatment failure and recurrence group was matched to successfully treated controls based on age, sex, and HIV status. Additionally, for the RNA sequencing analysis, matching also included Ralph scores, a metric for grading Chest X-ray severity in adults with smear-positive pulmonary TB[45]. In the following, we analyzed the differences in demographic and clinical characteristics between all participants in the TB treatment failure and recurrence group and the control group (Table 1). First, baseline C-reactive protein (CRP) levels were 1.7-fold higher in the control group than in the recurrence and treatment failure group (BL p = 0.018) (Table 1). As expected, treatment failure and recurrence cases exhibited a higher likelihood of testing culture positive for M2 (LC: p = <0.001, SC: p = 0.031) and M6 (LC: p = <0.001, SC: p = 0.025) compared to the control group. In addition, treatment failure and recurrent TB cases had higher sputum smear positivity grades at BL (p = 0.032), and the percentage of smear positive results at M6 was higher (p = 0.026) compared to the control group. No significant differences were found between the two groups if comparing X-ray results using the Ralph Score at M6 (Table 1). We also examined six common tuberculosis symptoms—cough, bloody cough, fever, weight loss, night sweats, and body weakness—at BL and M6. At BL, all TB treatment failure and recurrence cases had at least two symptoms, while 13% of controls had only one symptom (p = 0.037) (Table 1).

We further compared the individual treatment failure and recurrence subgroups—non-converters, reverters at EOT and recurrence after EOT—with the controls (Supplementary Table 2). Especially non-converters (n = 7) exhibited clinical differences compared to the successfully treated patients; however, due to the limited sample size, these findings should be interpreted with caution. This group had a consistently lower Body Mass Index (BMI) at BL (p = 0.011) and at M6 (p = 0.001) and exhibited more persistent lung damage, as defined by elevated Ralph scores at M6 (p = 0.019).

In contrast, the reverter at EOT subgroup (n = 14) showed no significant clinical differences compared to the control group, except for a lower CRP value at BL (p = 0.008) (Supplementary Table 2). Notably, 9 of 14 participants were MTB-positive only in liquid culture, not in solid culture or smear microscopy. Of these, 5 were either asymptomatic (n = 3) or had only one symptom (n = 2).

**Table 1 | Comparison of demographic and clinical characteristics between patients with TB treatment failure or recurrence and matched controls (by site, sex, age, HIV status, and additionally Ralph scores for RNAseq analysis)**

| Variables | Treatment failure and recurrence group n = 40 | Control group n = 37 | p-value |
|---|---|---|---|
| TAM-TB /RNAseq, n | 20/37 | 19/35 | |
| Treatment failure and recurrence group, n (%) (non-converter reverter at EOT recurrence after EOT) | 7 (17.5) 14 (35) 19(47.5) | - | - |
| **Matching Criteria** | | | |
| Site (Mzb/Tz/Gmb/Rsa), n | 2/18/10/10 | 3/17/9/8 | >0.98[a] |
| Sex (Male), n (%) | 27 (68) | 24 (65) | >0.81[b] |
| HIV positive at BL, n (%) | 14 (35) | 14 (38) | >0.8[b] |
| Age, median (IQR) | 36 (30–43) | 33 (30–42) | 0.93[c] |
| **BMI, median (IQR)** | | | |
| Baseline | 18.0 (16.8–19.3) | 18.9 (17.5–21.4) | 0.10[d] |
| Month 6 | 19.9 (17.8–21.1) | 21.0 (19.7–22.7) | 0.06[d] |
| CRP value BL, median (IQR) | 92.4 (59.2–121.3) | 161.0 (81.4–243.4) | 0.018[d] |
| **Microbiological results *** | | | |
| **Sputum smear positivity, n (%)** | | | |
| Screening/Baseline | 39 (97.5) | 26 (81) | 0.032[a] |
| Month 2 | 5 (14) | 1 (3) | 0.32[a] |
| Month 6 | 7 (18) | 0 (0) | 0.026[a] |
| **Liquid culture positivity, n (%)** | | | |
| Screening/Baseline | 40 (100) | 34 (100) | |
| Month 2 | 20 (59) | 5 (16) | <0.001[b] |
| Month 6 | 16 (44) | 0 (0) | <0.001[b] |
| **Solid culture positivity, n (%)** | | | |
| Screening/Baseline | 33 (85) | 28 (78) | 0.45[b] |
| Month 2 | 11 (31) | 3 (9) | 0.031[b] |
| Month 6 | 6 (17) | 0 (0) | 0.025[a] |
| **Chest X Ray - Ralph Score, median (IQR)** | | | |
| Month 6 | 10 (5–29) | 5 (5–29) | 0.24[c] |
| **Number of symptoms (Cough, bloody cough, fever, weight loss, night sweats, body weakness)** | | | |
| Baseline, ≥2/1/0, % | 100/0/0 | 87/13/0 | 0.037[a] |
| Month 6, ≥2/1/0, % | 13/33/54 | 14/32/54 | 0.67[a] |

Data were ≥85% complete for all variables except CRP, with 67% availability in both groups.
EOT end of treatment, Mzb Mozambique, Tz Tanzania, Gmb Gambia, Rsa Republic South Africa, BMI body mass index.
CRP: C-reactive protein, an unspecific inflammation marker.
Ralph Score: Radiographic score which considers the extent of lung infiltrate and the presence or absence of cavities.
* Percentages are based on valid results only; indeterminate samples (≤16% per timepoint) were excluded from the analysis.
[a]Fisher's exact test.
[b]Pearson's Chi-squared test.
[c]Wilcoxon rank sum test.
[d]Wilcoxon rank sum exact test. All tests were two-sided.

### Comparable decline of MTB-specific CD4 T cell activation during treatment in TB reversion, recurrence, and control groups

MTB125 was as sensitive as PPD in overnight in vitro restimulation assays for detecting MTB-specific CD4⁺ T cells via IFN-γ production. Notably, MTB125 induced earlier and more pronounced CD27 upregulation after treatment initiation compared to PPD (Supplementary Fig. 3). Given its defined protein composition and consistent performance, MTB125 was used for further analyses.

The dynamics of MTB-specific CD4 T-cell activation from treatment initiation until end of treatment (M6) were comparable between the successfully treated control group, reverters at EOT and recurrence cases after EOT (Fig. 2). In those groups, the activation marker CD38 was highest at BL and declined during treatment duration, with the most significant drop between BL and M2 (Fig. 2a). At M6, controls, reverters, and recurrence cases exhibited similar profiles, despite the detection of tuberculosis in the reverter subgroup at that time. The activation marker HLA-DR and Ki67 displayed similar longitudinal patterns to CD38 expression (Fig. 2b-c). In contrast, CD27 continuously increased throughout treatment, with significant changes between BL and M2 ($p = 0.01$) in the treatment failure and recurrence group and BL and M4 ($p = 0.035$) in the control group (Fig. 2d).

In the non-converter subgroup ($n = 4$) with persistent positive MTB culture or smear microscopy throughout TB treatment, MTB-specific CD4 T cell activation also initially declined significantly but remained high compared to other sub-groups, especially at M4. Notably, Ki67 was the most discriminating factor for the non-converter subgroup at both M2 and M4, compared to all other markers (Fig. 2c). Given the very small sample ($n = 4$), these results should be interpreted with caution, as limited statistical power may reduce reliability.

Generally, elevated CD38 levels were associated with positive microbiological results, except at EOT in the reverter subgroup, where the frequency of CD38 positivity was low despite positive microbiological findings (Fig. 3).

### Elevation of MTB-specific T cell activation identifies recurrent tuberculosis after end of treatment, but not reversion at end of treatment

We next focused our analysis on samples collected at the time of reversion and recurrence, respectively. Within the group subjected to TAM-TB assessment, TB reversion was detected in the sputum at M6 ($n = 7$), and TB recurrence occurred at M9 ($n = 3$), M12 ($n = 3$), M18 ($n = 2$), or M30 ($n = 1$) (total recurrence cases $n = 9$). For the following analysis, only recurrence cases with samples available at the recurrence timepoints M9 or M12 were included ($n = 5$) and compared with matched controls (M6: $n = 16$; M9/M12: $n = 9$). In recurrence cases (detected at M9 or M12 after treatment initiation), the MTB-specific T-cell markers were significantly higher in the recurrence group compared to the control group for CD38 ($p = 0.002$), HLA-DR ($p = 0.016$) and Ki67 ($p = 0.0023$), as shown in Fig. 4 a-c, right panel. The median frequency of CD38 + MTB-specific CD4 T cells was 58.8% in the recurrence group and 18.4% in the control group after EOT (Fig. 4a, right panel). Using our predefined cutoff of 31.6% of IFNγ+ CD38 + CD4 T cells, the recurrence cases were detected with a sensitivity of 100% (95% CI: 56.6–100%) and a specificity of 77.8% (95% CI: 56.5–99.4%) with an AUC of 0.98 (95% CI: 91–100%). No significant difference was detected in the expression of the maturation markers CD27 at M9 or M12 (CD27: $p = 0.7$, Fig. 4d, right panel). TAM-TB analysis of the few available samples prior to TB recurrence (M9 or M12) showed slightly elevated CD38 expression in the recurrence group (Supplementary Fig. 4a, right panel). Furthermore, three of four individuals had activated MTB-specific T cells when last assessed—between 3 and 21 months prior to the diagnosis of recurrence (Supplementary Fig. 4b). In contrast, when only considering cases with TB reversion at EOT no differences in T cell activation were observed (CD38 $p = 0.37$, HLA-DR $p = 0.082$, Ki67 $p = 0.58$; Fig. 4a-c, left panel). Likewise, when participants were still under treatment at the visit preceding TB reversion or recurrence, no differences in TB-specific activation markers were observed (Supplementary Fig. 4a, left panel). Only the

maturation marker CD27 showed a moderate difference at TB reversion ($p = 0.035$; Fig. 4d, left panel) and before reversion or recurrence during ongoing treatment ($p = 0.014$, Supplementary Fig. 4d), despite considerable overlap between the groups.

In summary, these findings suggest that the TAM-TB assay can reliably detect recurrent tuberculosis—potentially months before clinical or sputum-based diagnosis—while failing to identify subclinical reversion at the end of treatment.

### Transcriptomic signatures mirror MTB-specific CD4 T-cell activation patterns in TB non-converters, reverters, recurrence, and controls

Finally, we assessed the diagnostic and predictive potential of three parsimonious gene expression signatures with straightforward, pre-defined TB score calculations (MAMS6, RISK6, Sweeney3) for detecting TB treatment failure and recurrence[16,24]. We analyzed the temporal dynamics of these signatures during and after TB treatment in individuals with TB treatment failure and recurrence ($n = 37$) and matched controls ($n = 35$, Fig. 5a). Overall, similar to the trends observed during TAM-TB analysis, MAMS6 and RISK6 scores were elevated at baseline and declined following treatment initiation. Likewise, Sweeney3 scores reflected a positive treatment response in both groups. Notably, in participants who did not achieve culture-negative status ($n = 6$), the signature scores remained principally above (MAMS6 and RISK6) or below (Sweeney3) the global average, suggesting persistent immunoactivation in these participants.

None of the signatures reliably predicted TB treatment failure or recurrence at baseline (Fig. 5a). However, throughout follow-up, participants who later experienced TB treatment failure or recurrence generally exhibited either higher (MAMS6, RISK6) or lower (Sweeney3) TB risk scores than controls. Still, at M6, none of the RNA signatures differentiated reversion at EOT from controls (Fig. 5b, upper panel). Consistent with MTB-specific T-cell activation, transcriptomic signatures reliably detected TB recurrence happening at month 9 or later ($n = 12$ recurrence cases, $n = 15$ controls, Fig. 5b lower panel). Among the signatures, MAMS6 and RISK6 demonstrated the highest performance, achieving a sensitivity of 75% and specificity of 93% (95% CI: MAMS6 50–100% and 0–100%; RISK6 50–100% and 0–93%) (Youden's index). Sweeney3 exhibited the same sensitivity (75%) but lower specificities of 87% (95% CI: 50–100% and 0–100%). The AUCs were comparable across all signatures: MAMS6 (0.78; 95% CI 0.56-1), RISK6 (0.81; 95% CI 0.6–1), and Sweeney3 (0.83; 95% CI 0.66–1).

In summary, similar to the TAM-TB analysis, transcriptomic signatures effectively detect TB recurrence after treatment completion, while patients experiencing TB reversion at EOT remained largely indistinguishable from successfully treated patients.

### Level of MTB-specific CD4 T-cell activation correlates with transcriptomic signature scores across patient groups

We next analyzed correlations between the biomarkers examined in this study to assess their potential complementarity for TB outcome discrimination. Specifically, we compared TAM-TB results (%CD38+ of MTB-specific IFNγ + CD4 T cells) with three transcriptomic signature scores (Sweeney3, RISK6, and MAMS6) in patients with cured TB, treatment failure and TB recurrence. Analysis was restricted to individuals with paired transcriptomic and TAM-TB data at the same timepoint (controls $n = 16$, non-converters $n = 3$, reverter at EOT $n = 6$, recurrence after EOT $n = 7$).

Pearson's correlation analysis revealed significant associations between all transcriptomic signatures and TAM-TB (Fig. 6). Correlations among the transcriptomic signatures themselves were stronger (MAMS6~Sweeney3 $r = -0.93$; MAMS6 ~ RISK6 $r = 0.9$; RISK6~Sweeney3 $r = -0.77$) than between each signature and TAM-TB (MAMS6 ~ TAM-TB $r = 0.65$, RISK6 ~ TAM-TB $r = 0.64$, Sweeney3~TAM-TB $r = -0.58$). The lowest correlation was observed between Sweeney3 and TAM-TB (Fig. 6b). The low number of paired samples at the timepoint of recurrence after EOT

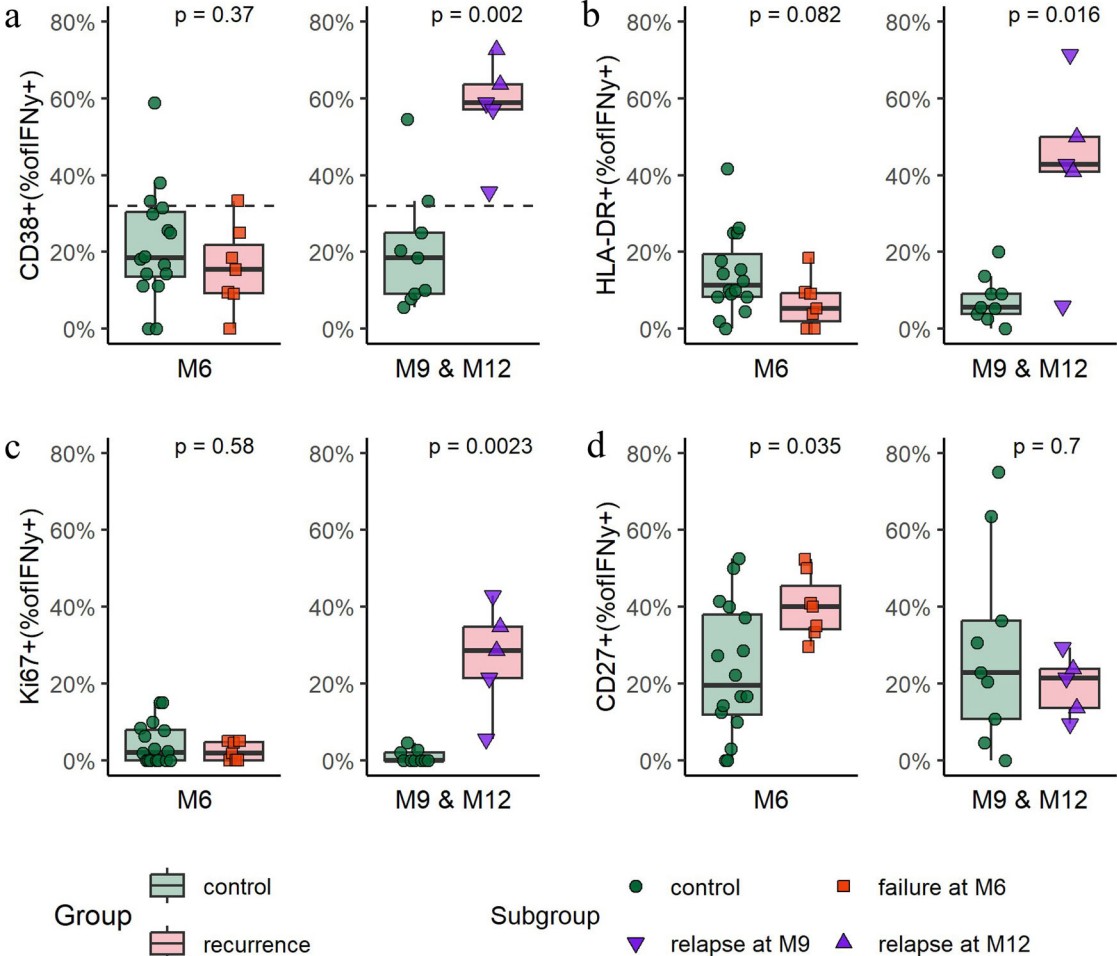

**Fig. 4 | Phenotypic profiles of MTB- specific CD4 + T cells at the time of TB reversion or recurrence compared to successfully treated controls.** Samples were collected at the time of reversion (M6; reverter at EOT, n = 7) or recurrence (M9 and M12; recurrence after EOT, n = 5), and analyzed for expression of CD38 (**a**), CD27 (**b**), Ki67 (**c**) and HLA-DR (**d**). Matched controls were included for comparison (M6, n = 16; M9/M12, n = 9). The x-axis indicates the time point of the second TB episode, and colors and shapes denote subgroups based on their timing. Dashed line in **a** indicates the cutoff for active tuberculosis, analyzed only for CD38. The box represents the interquartile range (IQR), with whiskers extending to the farthest points within 1.5 times the IQR. Statistical analysis was done using the Wilcoxon Rank-Sum Test (two-sided). *P*-values were calculated for all group comparisons, with $p \le 0.05$ considered statistically significant. EOT end of treatment, BL baseline, M month.

(n = 5; 2 recurrences, 3 controls) precluded a combined sensitivity analysis of TAM-TB and transcriptomic signatures.

Overall, these results indicate that while TAM-TB and transcriptomic signatures are moderately correlated, they may capture complementary aspects of host response, supporting their potential combined use for TB outcome prediction.

## Discussion

This study evaluated the potential of the sputum-independent TAM-TB assay and three TB RNA signatures to detect recurrent TB or treatment failure, compared to successfully treated controls with favorable TB outcomes. Both the TAM-TB and transcriptomic signatures reliably detected TB disease at baseline in controls, treatment failure and recurrence cases. Additionally, TB recurrence at the time of diagnosis was detected with high sensitivity and specificity. Both techniques also allowed for detection of non-sputum conversion. However, neither approach was able to predict subsequent TB recurrence already at baseline, nor did they effectively detect reversion at EOT.

Following treatment completion, the TAM-TB assay demonstrated high accuracy in detecting recurrent TB, achieving 100% sensitivity and 77.8% specificity. Curative treatment for pulmonary TB may not always completely eradicate *Mycobacterium tuberculosis*[46], potentially leaving residual viable bacilli. Thus, the MTB-specific T-cell activation dynamics observed in our study might reflect both treatment-induced reductions in bacterial load and subsequent bacterial regrowth in recurrent cases. This hypothesis is further supported by the detection of specific activation occurring 3–21 months prior to the diagnosis of recurrent TB. Our previous study in individuals living with HIV corroborates these findings; both incipient and recurrent TB were characterized by elevated MTB-specific T-cell activation both before and at the time of diagnosis, as well as during recurrence[21].

In participants who did not culture convert, a continuously activated TAM-TB profile contrasted with the mostly resolved activation profile observed by month 4 in cured TB. While T cell activation initially declined after treatment initiation also in non-converters, it mostly remained above the "active TB" threshold throughout TB treatment, suggestive of an only transient and insufficient bactericidal effect in these individuals. Increased MTB-specific T cell activation late in TB treatment is therefore likely a risk factor for treatment failure. The absent drug resistance measured at BL suggests that this was not the cause of non-conversion. We found no strong association between missed doses and treatment failure in the non-conversion subgroup, as suggested by Thompson et al.[33]. When examining adherence per dose, it was noted that 14% of doses were missed by non-converters compared to 9% of doses missed by cured patients. However, it is

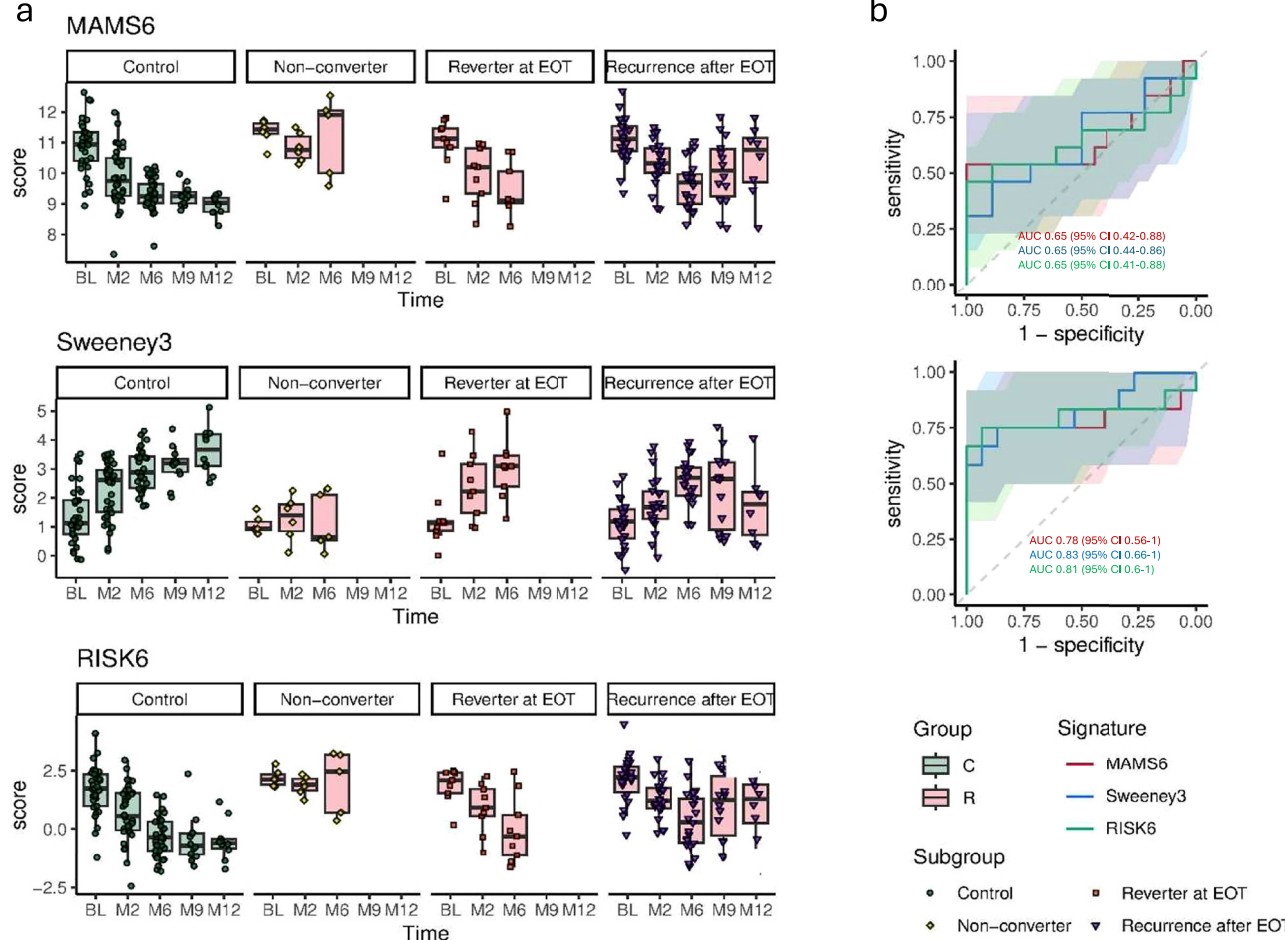

**Fig. 5 | Parsimonious transcriptomic signatures in patients with cured TB (Control Group) and those with recurrence after EOT or treatment failure (reverters at EOT and non-converters). a** Changes in signature scores over time are shown in months (0–12) for the Control Group (green, $n = 35$) and the TB treatment failure and recurrence group (red, $n = 37$). Colors and shape codes denote the group classifications based on the time point of the second episode, either during, at or after EOT. Patients are not included in the plot beyond their reversion or recurrence time point. The box represents the interquartile range (IQR), with whiskers extending to the farthest points within 1.5 times the IQR. **b** ROC curves for transcriptomic signature performance in detecting TB reversion or recurrence. Upper panel: ROC curve for M6 signature scores comparing participants with reversion at EOT to controls. Lower panel: ROC curve for M9 or M12 signature scores comparing participants with recurrence after EOT to controls. EOT end of treatment, *C* control group, *R* recurrence and treatment failure group (non-converter, reverter at EOT, recurrence after EOT).

noteworthy that non-converters shared multiple known risk factors for treatment failure including being male, having cavities or a Ralph score ≥50, low BMI, a baseline sputum-smear grade of 3+, and month 2 culture positivity[10].

In contrast to non-converters and recurrence cases after EOT, neither of the host biomarker diagnostic approaches detected reversion cases occurring at EOT. The unremarkable clinical presentation seen in these cases at this time point differed from the high-risk profile found in non-converters. TB reversion at EOT was detected in 9 of 14 patients by a single isolated liquid culture. Given that over half of these participants presented with no ($n = 3$) or only one symptom ($n = 2$), this raises possibility of false-positive diagnoses, as liquid culture can be prone to cross-contamination[47–49]. The reversion TB cases in this study might therefore consist of a mixed group of those with truly reversion of TB (clinical and subclinical) and others with false positive culture results. Yet, liquid culture remains the diagnostic gold standard for TB. The discovery of five false positive cases out of 1430 TB Sequel participants, accounting for a mere 0.35%, is still consistent with the high specificity of >99% achieved by this method[50].

The TAM-TB assay, using MTB125 synthetic peptides for in vitro T cell stimulation[51], effectively differentiated sputum-confirmed clinical TB before treatment initiation from cured TB at EOT using predefined CD38 expression thresholds. But no clear T cell marker pattern distinguished patients who experienced TB reversion or recurrence from those who did not during treatment. In both groups, MTB-specific T cell activation declined as expected during treatment[12,19,21]. While most reverters and recurrent TB cases also fell below the clinical TB threshold by month 4, several cured participants intermittently exceeded this threshold, suggesting that treatment may not always fully resolve TB[46]. Instead, reduced pathogen loads, through antibiotic treatment, may contribute to more efficient immune control or complete elimination of viable MTB after EOT. The similar dynamics of TAM-TB activation and maturation markers between TB reversion or recurrence and control groups during treatment suggest a.) similar levels of metabolically active bacteria and b.) comparable dynamics of bacterial clearance between these groups.

Transcriptomic risk scores followed similar dynamic patterns throughout the TB treatment, clearly differentiating recurrent TB from successfully treated TB after EOT. TB score profiles of non-converters also differed from controls, particularly at M6. However, consistent with the TAM-TB, transcriptomic signatures failed to detect reverters at EOT, which suggests that the disease burden in this group may be relatively low, which is consistent with previous research[16,33,52]. Interestingly, transcriptomic

**Fig. 6 | Correlation analysis of CD38 + MTB-specific IFNγ+ CD4 T cell frequencies (TAM-TB) and the transcriptomic signatures in patients with cured TB (control group) and those with treatment failure (non-converter and reverter at EOT) or recurrent TB.** Analysis was restricted to individuals with paired transcriptomic and TAM-TB data at the same timepoint. **a** Pearson correlation of the TAM-TB and transcriptomic signatures, with red indicating negative and blue indicating positive correlations. **b–d** Scatter plots of TAM-TB scores versus transcriptomic signature scores for individuals with paired transcriptomic and TAM-TB data: control ($n = 16$), non-converters ($n = 3$), reverter at EOT ($n = 6$), and recurrence after EOT ($n = 7$). All analyzed time points with paired data are shown. Each plot features a regression line (linear fit) with a 95% confidence interval, Pearson coefficient (R) and p-value are calculated separately (two-sided test). EOT end of treatment, TAM-TB T cell activation marker analysis).

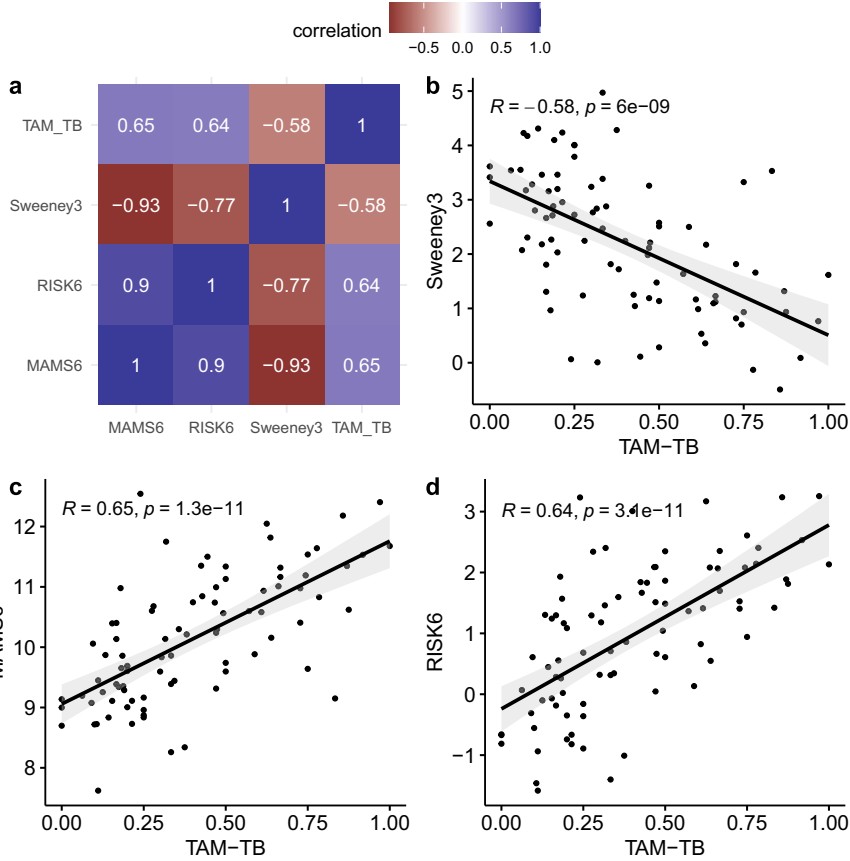

signatures at BL and M2 showed higher (Sweeney3) or lower (MAMS6 and RISK6) TB scores in the treatment failure and recurrence group compared to controls. This aligns with prior research indicating differential gene expression in recurrence already at the onset of treatment[33,35] and supports the notion of inherently different immune responses rather than a differential response to treatment[35]. Correlation analyses between transcriptomic signatures and TAM-TB showed moderate agreement, suggesting that combining these biomarkers may have the potential to improve diagnostic precision for TB outcomes.

Baseline CRP levels were higher in controls than in the TB treatment failure and recurrence group overall, with especially reverter at EOT showing significantly lower baseline values. This contrasts with Ronacher et al., who reported higher CRP in relapses and comparable baseline CRP levels in treatment failures (defined similarly to our reverter at EOT subgroup) and controls, possibly due to differences in cohort characteristics[53].

One limitation of this study is the relatively small sample size. Although the inclusion of 40 recurrence and TB treatment failure cases is substantial compared to prior studies, the heterogeneity within these cases required subgroup analyses, further reducing statistical power. Although both methods effectively identify recurrence after EOT, their capacity to predict or detect reversion during treatment remains limited. Our study also did not differentiate between TB reactivation and reinfection. We therefore cannot completely exclude the possibility that TB reinfection caused TB recurrence, particularly given the high TB incidence in the study regions. Moreover, cases of subclinical tuberculosis recurrence may have been missed, as sputum testing after EOT was only performed upon clinical suspicion of TB recurrence. In addition, since the TB Sequel study is observational rather than interventional, follow-up procedures reflect real-world clinical practice rather than the highly controlled conditions of a clinical trial. This may limit the sensitivity for detecting TB recurrence events and should be considered when interpreting "disease-free" outcomes among controls.

A key strength of this study is its thorough categorization of treatment failure and recurrence cases into three distinct subgroups: nonconverters, reversion at EOT, and recurrence after EOT, allowing for a more nuanced examination of clinical characteristics and biomarker performance. The analysis of two sputum-independent host biomarkers MTB-specific T-cell activation assays and transcriptomic signatures showed consistent results, enhancing conclusiveness of our study. Both MTB-specific T-cell activation and transcriptomic TB score profiles were identified as critical indicators of recurrent TB following treatment and non-culture conversion. Because the implementation of transcriptomic TB risk scores is less complex compared to the flow cytometry-based TAM-TB assay, the transcriptomic approach may be a more feasible sputum-independent alternative for clinical application in the field[54].

## Data availability
Source data underlying the main figures, including the TAM-TB cohort and the relevant clinical datasets, are provided as a single Excel workbook in the Supplementary Data (Supplementary Data 1). The workbook contains multiple sheets corresponding to the individual figures. RNA sequencing data will be deposited in the ENA repository (Accession number PRJEB101203).

## Code availability
The code for the analysis of the RNA signatures is provided at https://github.com/TropI-LMU/BauerEtAl2025.

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

## Acknowledgements

The authors would like to thank all colleagues and partners involved in the TB Sequel project for their dedication and invaluable contributions. Special thanks are extended to all study participants in South Africa, Gambia, Mozambique, and Tanzania, whose involvement made this project possible. The TB Sequel project is funded by the German Ministry for Education and Research (BMBF, 01KA1613) and is part of the Research Networks for Health Innovations in Sub-Saharan Africa. The experimental and data analytical work presented here was supported by the German Ministry for Education and Research through funding from the Deutsches Zentrum für Infektionsforschung (DZIF, TTU-TB personalized medicine TTU 02_813). The funders did not influence the study design, data collection, analysis, or interpretation; the writing of the manuscript; or the decision to submit the paper for publication. All authors confirm they had complete access to the study data and take full responsibility for the decision to submit the manuscript for publication.

## Author contributions

C.G., A.R., and K.H. were responsible for study development, funding acquisition, supervision of study conduct, and data analysis strategy. The setup and conduct of the clinical study, including data and sample collection, were carried out by M.R., S.C., J.S., A.K., M.C., N.N., and C.K. Laboratory data collection and analysis were performed by B.B., O.B., M.A., and A.B. B.B. and C.G. developed the paper and wrote the first draft. Critical review of the paper was performed by C.G., K.H., J.S. and A.R. All co-authors read, commented on, and approved the final version of the paper.

## Funding

## Competing interests

The authors (M.A., O.B., M.H., K.H., C.G.) have submitted a European patent application related to the MAMS_6 transcriptomic signature presented in the manuscript. The application is currently unpublished and pending. Otherwise, the authors have no competing interests.

## Additional information

[1]Institute of Infectious Diseases and Tropical Medicine, LMU University Hospital, LMU Munich, Munich, Germany. [2]German Center for Infection Research, Partner site Munich, Munich, Germany. [3]Fraunhofer Institute for Translational Medicine and Pharmacology ITMP, Immunology, Infection and Pandemic Research, Munich, Germany. [4]NIMR-Mbeya Medical Research Center (MMRC), Mbeya, Tanzania. [5]University of Dar es Salaam-Mbeya College of Health and Allied Sciences, Mbeya, Tanzania. [6]Instituto Nacional de Saúde, Marracuene, Maputo province, Mozambique. [7]Unit Global Health, Helmholtz Zentrum München, German Research Centre for Environmental Health (HMGU), Neuherberg, Germany. [8]Clinical HIV Research Unit, Helen Joseph Hospital, Department of Internal Medicine, University of the Witwatersrand, Johannesburg, South Africa. [9]School of Public Health, Faculty of Health Sciences, University of the Witwatersrand, Johannesburg, South Africa. [10]The Aurum Institute, Johannesburg, South Africa. [11]Vaccines and Immunity Theme, MRC Unit, The Gambia at LSHTM, Fajara, The Gambia. [18]These authors contributed equally: Kathrin Held, Andrea Rachow, Christof Geldmacher. ✉e-mail: bauer.be@campus.lmu.de; bauer.bernadette@web.de

**On behalf of the TB sequel consortium**

Beate Kampmann[12], Basil Sambou[12], Abi-Janet Riley[12], Binta Sarr[12], Caleb Muefong[12], Georgetta Daffeh[12], Olumuyiwa Owolabi[12], Ben Dowsing[12], Azeezat Sallahdeen[12], Shamanthi Jayasooriya[12], Abdou Sillah[12], Monica Davies[12], Alhaji Jobe[12], Momodou Jallow[12], Salieu Barry[12], Lamin Bah[12], Simon Badjie[12], Kairaba Kanyi[12], Gambia Sowe[12], Isatou Loum[12], Awa Touray[12], Mustapha Bah[12], Rohey Jallow[12], Simon Donkor[12], Issa Sabi[4], Tina Minja[4], Daniel Mapamba[4], Emmanuel Sichone[4], Lwitiho Sudi[4], Elimina Siyame[4], Julieth M. Lalashowi[4], Ian Sanne[13], Lyndel Singh[13], Jaclyn Bennet Denise Evans[13], Kamban Hirasen[13], Nelly Jinga[13], Ilesh Jani[14], Nilesh Bhatt[14], Sofia Viegas[14], Carla Madeira[14], Khalide Azam[14], Cláudio Abujate[14], Narciso Macie[14], Nádia Sitoe[14], Salomão Manjate[14], Vânia Maphossa[14], Alberto Machaze[14], Cristovão Matusse[14], Antonio Machiana[14], Candido Azize[14], Arlindo Machava[14], Celina Nhamuave[14], Elvira Monteiro[14], Olena Ivanova[1], Anna-Maria Mekota[1], Elmar Saathoff[1], Friedrich Riess[1], Fidelina Zekoll[1], Gavin Churchyard[10], Robert Wallis[10], Kavindhran Velen[10], Farzana Sathar[10], Fadzai Munedzimwe[10], Stefan Niemann[15], Matthias Merker[15], Viola Dreyer[15], Ulrich Schaible[15], Christoph Leschczyk[15], Lindsay Zurba[16] & Knut Lönnroth[17]

[12]Medical Research Council (MRC) Unit, The Gambia at LSHTM, Fajara, The Gambia. [13]WITS (University of Witwatersrand), Johannesburg, South Africa. [14]Instituto Nacional de Saúde (INS), Ministry of Health, Maputo, Mozambique. [15]Research Center Borstel, Leibniz-Center for Medicine and Biosciences (FZB), Borstel, Germany. [16]Education for Health Africa, Mount Edgecombe, South Africa. [17]Karolinska Institute, Stockholm, Sweden.

