## [Transparent Peer Review file · Communications Medicine]

Host response biomarkers of tuberculosis recurrence and treatment failure

Corresponding Author: Ms Bernadette Bauer

Version 0:

Reviewer comments:

Reviewer #1

(Remarks to the Author)

This study evaluates host-derived, sputum-independent biomarkers—namely MTB-specific T-cell activation (TAM-TB) and transcriptomic signatures (MAMS6, RISK6, Sweeney3)—for detecting TB treatment failure and post-treatment recurrence. Given the limitations of sputum smear and culture, especially in individuals unable to produce sputum, the development of reliable non-sputum biomarkers is a pressing and unmet need. The authors employ novel and complementary approaches to address this gap, and the findings are compelling. However, the study is limited by acknowledged small sample sizes, especially in key subgroups, and by the lack of molecular speciation to distinguish relapse from reinfection. This is an important study worthy of publication in this journal, with potential to influence TB monitoring strategies, particularly where sputum-based tools fall short. However, major issues, especially regarding terminology, clarity, consistency, methodological transparency, and overinterpretation, must be addressed. I recommend major revisions prior to consideration for publication.

Major comments

1. The manuscript refers to treatment failures (e.g., sputum reverters and non-converters) as “recurrence at the end of treatment,” which is incorrect.
“...cases were categorized into three subgroups: (a) Non-converters... (b) recurrence at the end of treatment... (c) recurrence after EOT.”
Terminology in the manuscript are used inconsistently, which makes sections of the manuscript challenging to understand. Please align terminology with WHO definitions: use “treatment failure” for non-converters and reverters, and “recurrence” only for events after EOT (relapse or reinfection). This should be applied consistently throughout the text, figures, tables, and abstract.
2. “...relapses after EOT, had negative microbiological results at EOT but developed signs of TB recurrence and positive microbiological results after EOT...”. Without genotyping or sequencing, it is not possible to distinguish whether recurrence cases after EOT represent relapse or new infections (as acknowledged in your discussion). Please avoid using the term “relapse” in the text/figures unless strain identity is confirmed. Rather use TB recurrence throughout. Please also clarify in the methods your definition of the recurrence endpoint (i.e., may be relapse or reinfection).
3. The abstract omits subgroup sizes, specific assays, sampling/measurement timepoints, and confidence intervals for diagnostic estimates. The abstract should specify the study setting, biomarker types (Which MTB-specific T-cell activation assays and transcriptomic signatures were measured?), sampling windows (e.g., M0, M6, M12), and 95% CIs for sensitivity/specificity/AUC values.
4. In the introduction, the statement “...the global success rate for first-line regimens remains suboptimal at 88%, partly due to the lengthy duration of therapy...” does not have a reference. Also, I don’t think that the statement is necessarily correct. There are multiple reasons for treatment failure; poor adherence due to lengthy duration of treatment is just one.
5. “Identification of TB patients with a high likelihood of treatment failure and disease recurrence will hence help reduce drug resistance...”. What is your hypothesis for reducing drug resistance. Also add a reference please.
6. Sample size is a major limitation (as acknowledged in the discussion), but especially for subgroup analyses. While the total recurrence group includes 40 participants, subgroup sizes are much smaller and further reduced by missing samples. “Especially non-converters (n=7) exhibited clinical differences...”
“In the non-converter subgroup (n=4)....”
Results from such small groups should be interpreted cautiously and the limited power explicitly stated when comparing subgroups. Be cautious over interpreting statistical tests/p-values.
7. Several methods are under-explained or ambiguous in the methods section. For example, it is not stated in methods how

transcriptomic signatures were calculated. According to the methods reported in the original papers? E.g., for Sweeney3, KLF2 -mean(GBP5+DUSP3), and for RISK6, the geometric mean of 3 downregulated genes – the geometric mean of the 3 upregulated genes? (or the pair-wise ensemble model)?

8. Please clarify in the methods at which timepoints samples were collected (PBMC, PAXgene, sputum).

9. “After the M6 follow-up, microbiological testing was only resumed if a recurrence of TB was suspected...”. No sputum collected at end of follow-up may have resulted in missed subclinical recurrences. Please discuss this limitation explicitly—it affects interpretation of “disease-free” outcomes among controls.

10. The selected p-value threshold may be inappropriate given low power. Consider whether using a strict $p \leq 0.05$ threshold is appropriate.

11. In Figure 1, under “excluded n=10, TAM-TB n=30, RNA-seq n=13 missing samples”. It is unclear how 10 participants could result in 30 missing samples. Same issue for the control group.

12. Table 1 is very difficult to interpret as it is packed with information/too busy (I don't like the n/n/n/n format – very challenging to read). I would suggest just providing just the baseline (and/or M6) characteristics of the groups, rather than multiple timepoints. You could then include n (%) for each variable.

13. Some figures omit p-values or confidence intervals. For example, in Figure 2 only some p values shown? Are the other p values above a certain threshold? Please provide p-values consistently across figures or clarify if nonsignificant. Include 95% confidence intervals for all sensitivity/specificity/AUC metrics in text and figures.

14. Suggest also stratifying figure/using facet plots for Figure 2 to look at controls vs the 3 subgroups (non-converters, reverters, and recurrence).

15. Figure 5B: Add 95%CI to AUC estimates.

16. MTB125 was as sensitive as PPD for detecting MTB-specific T cells after overnight in vitro restimulation and assessing their activation profile throughout TB treatment (Supplementary Figure 3).” What do you mean by “... was as sensitive...”. Unclear. Figure F also clearly shows differences between MTB125 and PPD.

17. “Notably, in participants who did not achieve culture negative status (n=6), the signature scores remained principally above (MAMS6 and RISK6) or below (Sweeney3) the global average...”. Difficult to see this pattern in the jumble of data point in Figure 5A. Please stratify/facet plot Figure 5A by non-converters, reverters, and recurrence subgroups.

18. “In recurrent cases, median MTB-specific activation was notably higher for diagnoses made at month 12 compared to those at month 9. This finding may indicate a progressively worsening disease state and increasing pathogen loads following treatment cessation, which increases MTB-specific T-cell activation. TB.” Overstatement. Very few datapoints, and you don't actually test this hypothesis.

Minor comments

1. “...who cannot produce sputum, including children and people living with HIV with paucibacillary disease...” These are distinct concepts (1. Inability to produce sputum and 2. Paucibacillary disease) and should not be presented as equivalent. Please rephrase.

2. “...before (re)occurrence of MTB bacilli in sputum.”

Consider rewording—this phrasing is awkward and ambiguous.

3. “Transcriptomic profiling has emerged as another powerful tool not only for diagnosing TB infection...”

Transcriptomic signatures do not diagnose latent TB infection. This should be corrected to “TB disease.”

4. “Baseline (BL) visit...”

Clarify whether BL corresponds to treatment initiation.

5. In the introduction, states “follows TB patients for 24 months after microbiological diagnosis”. But in methods, there is a month 30 visit... and later mentions “...mean follow-up of 35 months...”

6. “Recurrence was identified either within 6 months of treatment or after end of treatment (EOT) during the follow-up period.” Please clarify: “within 6 months of treatment” is ambiguous—do you mean during treatment?

7. “For the signatures that collapsed gene CT value into a score, we calculated the respective number using RNA-seq gene counts. While that makes any cut off value inappropriate, we expect that the trend and the degree of separation between groups remains unchanged.”

Please rephrase—this section is unclear and may confuse readers unfamiliar with CT-to-count mapping issues.

8. “Paired Wilcoxon rank-sum test...”

The correct name is “Wilcoxon signed-rank test” for paired comparisons.

9. “...baseline C-reactive protein (CRP) levels were slightly higher...”

This downplays the result; CRP was 1.7× higher. Please revise.

10. “Results: What does LC and SC stand for?”

Please define these terms on first use.

Reviewer #2

(Remarks to the Author)

The identification of reliable biomarkers for tuberculosis represents a critical advancement in the global effort to fight TB. Currently, microbiological tests such as culture remain the gold standard for monitoring treatment response; however, they are time-consuming, resource-intensive, and often fail to capture the complete clinical picture. Biomarkers offer the potential for a faster, more sensitive, and patient-centered approach to assess treatment efficacy and predict relapse risk. This is particularly important given the limitations of current tools in detecting viable bacilli and distinguishing between true cure and persistent subclinical disease. The development and validation of such biomarkers would not only enhance individualized patient care but also strengthen public health strategies aimed at reducing TB recurrence and transmission. This study proposes non-sputum biomarkers for predicting TB recurrence after treatment completion. Nevertheless, future validation in a larger cohort is required to confirm these findings. Importantly, the study provides a valuable foundation for future research

aimed at improving post-treatment monitoring and long-term outcomes in TB patients.

Major Concerns:

1. Clarification of MTB125

Please provide a clear explanation of what MTB125 refers to. Which specific peptides does it include?

2. Inconsistency in Methods and Results

The Methods section mentions stimulation with ESAT6/CFP10, yet no corresponding data or results are presented in the manuscript. Could you clarify whether this analysis was performed and, if so, why the results are not included?

3. Statistical Analysis

Have you considered applying a univariate logistic regression to assess the association between individual variables and the binary outcome (e.g., relapse/no relapse)? Furthermore, a multivariate logistic regression could be valuable to evaluate the simultaneous effect of multiple independent variables on the outcome.

4. Correlation Between Assays

Did you consider assessing the correlation between the TAM-TB assay results and RNA signatures across the different study groups? Such an analysis could strengthen the interpretation of the findings.

Version 1:

Reviewer comments:

Reviewer #1

(Remarks to the Author)

Congratulations on an excellent revision and a thorough, thoughtful rebuttal. Your responses comprehensively address my prior concerns and considerably strengthen the manuscript. I have only two minor follow-ups:

(1) there's a small typo in the Statistical Methods—please change “Walt” to “Wald”; and

(2) for data availability, please deposit the RNA-seq data in a public, field-appropriate repository and update the Data Availability statement with the accession IDs; likewise, please provide a public link/DOI for the analysis code.

Dr. Simon C. Mendelsohn

Senior Researcher

South African Tuberculosis Vaccine Initiative (SATVI), University of Cape Town

Reviewer #2

(Remarks to the Author)

The authors have satisfactorily addressed the reviewers' comments and provided thorough justifications in cases where the requested analyses could not be performed.

Response to Reviewers

Manuscript Title: Host response biomarkers of tuberculosis recurrence and treatment failure

Manuscript ID: COMMSMED-25-1036

Journal: Communications Medicine

Date: 5 September 2025

Dear Editors and Reviewers,

We sincerely thank the reviewers for their thoughtful and constructive comments. We have carefully addressed each comment and revised the manuscript accordingly. All changes are highlighted in yellow for your convenience. In addition, we have uploaded a clean version of the manuscript without highlighting it. Our point-by-point responses are detailed below.

We thank you again for taking the time to review our revised manuscript.

Bernadette Bauer, corresponding author, on behalf of all authors

RESPONSE to Reviewer 1: We sincerely thank the reviewer for the positive feedback on our study, the thorough review, and the thoughtful suggestions that helped improve our manuscript. We believe that these changes have strengthened the article considerably.

In particular, the adjustments in nomenclature as well as the suggestions regarding tables and figures have greatly enhanced the visualization and overall clarity of our manuscript.

Please find below our detailed responses to each point raised.

Major comments

1. The manuscript refers to treatment failures (e.g., sputum reverters and non-converters) as “recurrence at the end of treatment,” which is incorrect.

“...cases were categorized into three subgroups: (a) Non-converters... (b) recurrence at the end of treatment... (c) recurrence after EOT.”

Terminology in the manuscript are used inconsistently, which makes sections of the manuscript challenging to understand. Please align terminology with WHO definitions: use “treatment failure” for non-converters and reverters, and “recurrence” only for events after EOT (relapse or reinfection). This should be applied consistently throughout the text, figures, tables, and abstract.

RESPONSE:

Thank you for your insightful comment. Indeed, we carefully considered how best to define our subgroups when preparing the manuscript, and we appreciate the suggestion to adhere to WHO definitions.

Accordingly, we have revised the manuscript and updated the figures based on the updated WHO definitions for TB treatment outcomes (WHO, 2021; Eur Respir J 2021; 58: 2100804):

- **“Treatment failure”** now includes both **non-converters** (patients without culture conversion) and **“reverters at EOT”** (those who initially converted but then reverted to positive sputum at the end of the continuation phase).
- **“Reverters at EOT”** refers to a subgroup of WHO-defined reverters. While the WHO uses the term more broadly, we specify “reverters at EOT” to indicate that in our cohort, all reversions occurred at Month 6, corresponding to the end of the continuation phase.
- **“Recurrence”** is used exclusively for events occurring **after end-of-treatment (EOT)** and refers to relapse or reinfection, consistent with the WHO framework for post-treatment recurrence

We believe this clarification aligns our work with standardized definitions and enhances interpretability across the manuscript.

2. “...relapses after EOT, had negative microbiological results at EOT but developed signs of TB recurrence and positive microbiological results after EOT...”. Without genotyping or sequencing, it is not possible to distinguish whether recurrence cases after EOT represent relapse or new infections (as acknowledged in your discussion). Please avoid using the term “relapse” in the text/figures unless strain identity is confirmed. Rather use TB recurrence throughout. Please also clarify in the methods your definition of the recurrence endpoint (i.e., may be relapse or reinfection).

RESPONSE: We fully agree with the reviewer’s comment and have revised the manuscript accordingly, replacing all instances of “relapse” with “recurrence.” The definition of the

recurrence endpoint has also been clarified in **the Methods and Materials section, in the last paragraph of Study Participants and Sample Selection**, to indicate that it may represent either relapse or reinfection.

Additionally we added this to the **limitation section (page 13, paragraph1)** of our study: “Our study also did not differentiate between TB reactivation and reinfection. We therefore cannot completely exclude the possibility that TB reinfection caused TB recurrence, particularly given the high TB incidence in the study regions.

3. The abstract omits subgroup sizes, specific assays, sampling/measurement timepoints, and confidence intervals for diagnostic estimates. The abstract should specify the study setting, biomarker types (Which MTB-specific T-cell activation assays and transcriptomic signatures were measured?), sampling windows (e.g., M0, M6, M12), and 95% CIs for sensitivity/specificity/AUC values.

RESPONSE: We have adapted the abstract according to the reviewer’s suggestions and added a specified study setting, biomarker types, sampling windows as well as 95% CIs for sensitivity/specificity/AUC values.

4. In the introduction, the statement “...the global success rate for first-line regimens remains suboptimal at 88%, partly due to the lengthy duration of therapy...” does not have a reference. Also, I don’t think that the statement is necessarily correct. There are multiple reasons for treatment failure; poor adherence due to lengthy duration of treatment is just one.

RESPONSE: This statement (“...first-line regimens remains suboptimal at 88%...”) is derived from the WHO tuberculosis report 2024. We have additionally supplemented the reasoning to clarify that there are multiple reasons for treatment failure.

We have now changed the sentence in the **first paragraph of the introduction** as follows:

“Despite the availability of treatment, the global success rate for first-line regimens remains suboptimal at 88% (1), for a variety of reasons such as the lengthy duration of therapy, side effects and poor tolerability (2).”

5. “Identification of TB patients with a high likelihood of treatment failure and disease recurrence will hence help reduce drug resistance...”. What is your hypothesis for reducing drug resistance. Also add a reference please.

RESPONSE: We can hypothesize that identification of patients at risk of treatment failure allows for regimen adjustments thereby stopping ongoing bacillary replication under suboptimal therapy that might select for resistant TB strains. Also, for early detection of relapses with resistant TB: they may unknowingly transmit resistant strains to others.

We have now changed the sentence in the **first paragraph of the introduction** as follows:

“Identification of TB patients with a high likelihood of treatment failure and disease recurrence will hence help reduce drug resistance by allowing for treatment adjustments stopping ongoing bacillary replication under suboptimal therapy that might select for resistant TB strains (2, 4, 5)”

References:

1. Companion Handbook to the WHO Guidelines for the Programmatic Management of Drug-Resistant Tuberculosis. <https://www.ncbi.nlm.nih.gov/books/NBK247445/>
2. Multidrug resistance after inappropriate tuberculosis treatment: a meta-analysis; Marieke J. van der Werf; European Respiratory Journal 2012 39(6): 1511-1519; DOI: <https://doi.org/10.1183/09031936.00125711>
3. Determinants of drug-resistant tuberculosis: analysis of 11 countries; Espinal; <https://pubmed.ncbi.nlm.nih.gov/11605880/>

6. Sample size is a major limitation (as acknowledged in the discussion), but especially for subgroup analyses. While the total recurrence group includes 40 participants, subgroup sizes are much smaller and further reduced by missing samples.

“Especially non-converters (n=7) exhibited clinical differences...”

“In the non-converter subgroup (n=4)...”

Results from such small groups should be interpreted cautiously and the limited power explicitly stated when comparing subgroups. Be cautious over interpreting statistical tests/p-values.

RESPONSE: We have now emphasized the limitation of small sub-group analysis

Revised sentences: page 8, second paragraph, results/ Demographic and clinical characteristics of study participants section.

1. “Especially non-converters (n=7) exhibited clinical differences compared to the successfully treated patients; however, due to the limited sample size, these findings should be interpreted with caution.”

and page 9, second paragraph, result section

2. “In the non-converter subgroup (n = 4) with persistent positive MTB culture or smear microscopy throughout TB treatment, MTB-specific CD4 T cell activation initially declined significantly but remained high compared to other sub-groups, especially at M4. Notably, Ki67 was the most discriminating factor for the non-converter subgroup at both M2 and M4, compared to all other markers (Figure 2C). Given the very small sample (n = 4), these results should be interpreted with caution, as limited statistical power may reduce reliability.”

7. Several methods are under-explained or ambiguous in the methods section. For example, it is not stated in methods how transcriptomic signatures were calculated. According to the methods reported in the original papers? E.g., for Sweeney3, $KLF2 -\text{mean}(\text{GBP5}+\text{DUSP3})$, and for RISK6, the geometric mean of 3 downregulated genes – the geometric mean of the 3 upregulated genes? (or the pair-wise ensemble model)?

RESPONSE: We have now modified and added further explanations to the RNA processing, sequencing and signature calculation section as well as added a supplementary table describing signature score calculations.

Revised paragraph (pages 6-7, second paragraph, *Methods and Materials IRNA processing, sequencing and signature calculation*)

“The calculation was performed using the formulas and gene IDs in Supplementary Table 1; gene names in the formula represent log normalized counts. No significance testing was performed due to low numbers of subgroups. Performance of the signature was determined using Youden’s J statistic. AUCs were calculated using pROC package (1·18·5) (34).

Published cut-offs from the three signatures were established using qPCR derived CT values, hence these cut offs were not applicable to RNA sequencing data. Nonetheless we expect that the signatures will show a degree of separation between the groups when calculated from RNA sequencing and microarray data comparable to qPCR. Additional packages used for data handling were tidyverse (v. 2.0.0) and vroom (v. 1.6.5) packages. “

And added the Supplementary Table 1

Signature	Calculation	Symbol to ID mapping
Sweeney 3	$KLF2 - (GBP5 + DUSP3)/2$	KLF2 : ENSG00000127528 GBP5 : ENSG00000154451 DUSP3 : ENSG00000108861
RISK 6	$\sqrt[3]{GBP2 * FCGR1B * SERPING1} - \sqrt[3]{TUBGCP6 * TRMT2A * SDR39U1}$	GBP2 : ENSG00000162645 FCGR1B : FCGR1BP : ENSG00000198019 SERPING1 : ENSG00000149131 TUBGCP6 : ENSG00000128159 TRMT2A : ENSG00000099899 SDR39U1 : ENSG00000100445
MAMS 6	$mean(AIM2, FCGR1B, CD274, GBP1, SMARCD3, FLVCR2)$	AIM2 : ENSG00000163568 FCGR1B : FCGR1BP : ENSG00000198019 CD274 : ENSG00000120217 GBP1 : ENSG00000117228 SMARCD3 : ENSG00000082014 FLVCR2 : ENSG00000119686

8. Please clarify in the methods at which timepoints samples were collected (PBMC, PAXgene, sputum).

RESPONSE: We have incorporated the time points on **page 4, first paragraph, methods section** as follows:

“Sputum samples for microbiological testing were routinely collected at screening, BL, M2, M4, and M6. After the M6 follow-up, sputum testing was only performed if TB recurrence was clinically suspected based on signs and symptoms. Peripheral blood mononuclear cells (PBMCs) and PAXgene blood samples were collected at BL, M2, M4, M6, M9, M12, and in cases of suspected recurrence.”

9. “After the M6 follow-up, microbiological testing was only resumed if a recurrence of TB was suspected...”. No sputum collected at end of follow-up may have resulted in missed subclinical recurrences. Please discuss this limitation explicitly—it affects interpretation of “disease-free” outcomes among controls.

RESPONSE: We have now added this limitation explicitly in **the limitation section (page 13, paragraph 1)** as follows:

“In addition, since the TB Sequel study is observational rather than interventional, follow-up procedures reflect real-world clinical practice rather than the highly controlled conditions of a

clinical trial. This may limit the sensitivity for detecting TB recurrence events and should be considered when interpreting “disease-free” outcomes among controls.”

10. The selected p-value threshold may be inappropriate given low power. Consider whether using a strict $p \leq 0.05$ threshold is appropriate.

RESPONSE: We appreciate the reviewer’s comment regarding the p-value threshold. We applied a consistent significance threshold of $p \leq 0.05$ throughout the analysis, as stated in the Statistical Methods section. To improve clarity and avoid confusion, we have reworded the relevant text to explicitly state: “A p -value ≤ 0.05 was considered significant.”

We acknowledge that with limited power, interpretation of p-values should be cautious; however, we chose this conventional threshold to maintain consistency with similar studies.

11. In Figure 1, under “excluded n=10, TAM-TB n=30, RNA-seq n=13 missing samples”. It is unclear how 10 participants could result in 30 missing samples. Same issue for the control group.

RESPONSE: For clarification, we have added the following note to the **figure legend of the flowchart (Figure 1):**

Note: Exclusions were due to missing or poor-quality samples. Many missing samples in the TAM-TB analysis resulted from shipping issues and insufficient PBMCs in South Africa.

12. Table 1 is very difficult to interpret as it is packed with information/too busy (I don’t like the n/n/n/n/n format – very challenging to read). I would suggest just providing just the baseline (and/or M6) characteristics of the groups, rather than multiple timepoints. You could then include n (%) for each variable.

RESPONSE: We acknowledge that the original format of Table 1 may have been overly complex and challenging to interpret. We have now revised Table 1 as well as Supplementary Table 2. We have reformatted the data to display counts for culture and smear positivity and percentages (n (%)) for each variable, eliminating the previous n/n/n/n/n format. We believe these changes enhance the clarity and readability of the table while retaining all essential information.

13. Some figures omit p-values or confidence intervals. For example, in Figure 2 only some p values shown? Are the other p values above a certain threshold? Please provide p-values consistently across figures or clarify if nonsignificant. Include 95% confidence intervals for all sensitivity/specificity/AUC metrics in text and figures.

RESPONSE:

We appreciate the reviewer’s suggestion to provide p-values and confidence intervals consistently. We have carefully revised all relevant figures and text sections to include p-values uniformly. In figures where p-values are omitted, these correspond to nonsignificant results exceeding the significance threshold, and this is now explicitly clarified in the figure legends. Moreover, we have added 95% confidence intervals for all sensitivity, specificity, and AUC metrics throughout the manuscript and figures. These additions improve clarity and allow better assessment of the precision of our estimates.

14. Suggest also stratifying figure/using facet plots for Figure 2 to look at controls vs the 3 subgroups (non-converters, reverters, and recurrence).

RESPONSE: We thank the reviewer for this valuable suggestion. Accordingly, we have revised Figure 2 by stratifying the data into controls and the three subgroups (non-converters, reverters, and recurrence) using facet plots. This has greatly improved the clarity and interpretability of the figure.

15. Figure 5B: Add 95%CI to AUC estimates.

RESPONSE: We now have added 95% CI to Figure 5B.

16. MTB125 was as sensitive as PPD for detecting MTB-specific T cells after overnight in vitro restimulation and assessing their activation profile throughout TB treatment (Supplementary Figure 3).” What do you mean by “... was as sensitive...”. Unclear. Figure F also clearly shows differences between MTB125 and PPD.

RESPONSE: Thank you for pointing out this confusion. We meant that MTB-125 is as sensitive as PPD during vitro restimulation experiments in detecting MTB-specific CD4 T cells via their IFN γ production to then analyse their activation and maturation profile. Interestingly CD27 expression dynamics differed between MTB125 and PPD restimulated MTB-specific T cells after treatment initiation. MTB125-stimulated MTB-specific T cells appeared to reflect a treatment response by upregulation of CD27 compared to baseline, while PPD restimulated MTB-specific T cells remained largely CD27 negative.

To make this clearer we changed our wording on **page 8 paragraph 4:**

“MTB125 was as sensitive as PPD in overnight in vitro restimulation assays for detecting MTB-specific CD4⁺ T cells via IFN- γ production. Notably, MTB125 induced earlier and more pronounced CD27 upregulation after treatment initiation compared to PPD (Supplementary Figure 3). Given its defined protein composition and consistent performance, MTB125 was used for further analyses. “

17. “Notably, in participants who did not achieve culture negative status (n=6), the signature scores remained principally above (MAMS6 and RISK6) or below (Sweeney3) the global average...”. Difficult to see this pattern in the jumble of data point in Figure 5A. Please stratify/facet plot Figure 5A by non-converters, reverters, and recurrence subgroups.

RESPONSE: We thank the reviewer for this valuable suggestion. In response, we have updated Figure 5A by splitting the data into controls and the three subgroups (non-converters, reverters, and recurrence) using facet plots. This modification has substantially improved the clarity and interpretability of the figure.

18. “In recurrent cases, median MTB-specific activation was notably higher for diagnoses made at month 12 compared to those at month 9. This finding may indicate a progressively worsening disease state and increasing pathogen loads following treatment cessation, which increases MTB-specific T-cell activation. TB.” Overstatement. Very few datapoints, and you don't actually test this hypothesis.

RESPONSE: Thank you for your valuable feedback. Given the limited data and the speculative nature of the statement, we have decided to remove the sentence from the discussion to avoid overinterpretation.

Minor comments

1. "...who cannot produce sputum, including children and people living with HIV with paucibacillary disease..."

These are distinct concepts (1. Inability to produce sputum and 2. Paucibacillary disease) and should not be presented as equivalent. Please rephrase.

RESPONSE: We rephrased the sentence to clearly distinguish between inability to produce sputum and paucibacillary disease.

Revised sentence (page 3, paragraph 2, introduction section): "Moreover, sputum-based diagnostics have limitations, particularly in individuals with extrapulmonary TB, those unable to produce sputum (e.g., children), and patients with paucibacillary disease."

2. "...before (re)occurrence of MTB bacilli in sputum."

Consider rewording—this phrasing is awkward and ambiguous.

RESPONSE:

Revised sentence (page 3, paragraph 3, introduction section): "Furthermore, after treatment initiation, the decline in MTB-specific T-cell activation correlates with time to culture conversion. Moreover, such activated T cells can often be detected even before symptom onset and before MTB bacilli become detectable in sputum."

3. "Transcriptomic profiling has emerged as another powerful tool not only for diagnosing TB infection..."

Transcriptomic signatures do not diagnose latent TB infection. This should be corrected to "TB disease."

RESPONSE: We appreciate this clarification and have revised the sentence accordingly to "diagnosing TB disease."

4. "Baseline (BL) visit..."

Clarify whether BL corresponds to treatment initiation.

RESPONSE:

Revised sentence (page 4, paragraph 1 of methods and materials/ Study participants and sample selection): "The TB Sequel study followed TB patients through multiple visits: screening (days -6 to -1); baseline (BL), defined as the day of treatment initiation or up to 7 days afterward; day 14; months (M) 2, 4, and 6; and post-treatment visits at M9, M12, M18, M24, and M30 after enrollment."

5. In the introduction, states "follows TB patients for 24 months after microbiological diagnosis". But in methods, there is a month 30 visit... and later mentions "...mean follow-up of 35 months..."

RESPONSE: We agree that the original phrasing was unclear. TB Sequel includes an intensive follow-up period of 24 months after microbiological diagnosis, followed by a prolonged follow-up phase in which participants are scheduled to return to the study clinic every 6 months (± 2

months) until the end of the project. The total project duration is 5 years, with the final visit planned 3 months before project completion. We have clarified this in the Methods section as follows:

Revised sentences (page 3, paragraph 5, introduction section): “The cohort follows TB patients intensively for 24 months after microbiological diagnosis (31), with extended follow-up every 6 months thereafter for up to 5 years.”

And page 4, paragraph 1, Methods and Materials/Study participants and sample selection: “The TB Sequel study followed TB patients through multiple visits: screening (days –6 to –1); baseline (BL), defined as the day of treatment initiation or up to 7 days afterward; day 14; months (M) 2, 4, and 6; and post-treatment visits at M9, M12, M18 and M24 after enrollment, with additional follow-up every 6 months for up to 5 years.”

6. “Recurrence was identified either within 6 months of treatment or after end of treatment (EOT) during the follow-up period.”

Please clarify: “within 6 months of treatment” is ambiguous—do you mean during treatment?

RESPONSE:

Revised sentences (page 4, paragraph 3, Methods and Materials/Study participants and sample selection section):

“Cases of treatment failure and recurrence were identified either during the 6-month treatment period or after treatment completion (end of treatment, EOT) during the follow-up phase.”

And page 7, paragraph 1, Results/ Demographic and clinical characteristics of study participants section:

“Cases were classified as treatment failure if occurring during treatment, or as recurrence if identified after EOT.”

7. “For the signatures that collapsed gene CT value into a score, we calculated the respective number using RNA-seq gene counts. While that makes any cut off value inappropriate, we expect that the trend and the degree of separation between groups remains unchanged.”

Please rephrase—this section is unclear and may confuse readers unfamiliar with CT-to-count mapping issues.

RESPONSE: Thank you for highlighting the unclear wording. It has been addressed in the revised version of the text.

Revised sentences: Page 6-7, paragraph 2, Results/RNA processing, sequencing and signature calculation section

“Published cut-offs from the three signatures were established using qPCR derived CT values, hence these cut offs were not applicable to RNA sequencing data. Nonetheless we expect that the signatures will show a degree of separation between the groups when calculated from RNA sequencing and microarray data comparable to qPCR. Additional packages used for data handling were tidyverse (v. 2.0.0) and vroom (v. 1.6.5) packages.”

8. “Paired Wilcoxon rank-sum test...”

The correct name is “Wilcoxon signed-rank test” for paired comparisons.

RESPONSE: Thank you for pointing this out. We have corrected the terminology throughout the manuscript.

9. "...baseline C-reactive protein (CRP) levels were slightly higher..."
This downplays the result; CRP was 1.7× higher. Please revise.

RESPONSE: Revised sentences page 7, paragraph 1, Results section:

"First, baseline C-reactive protein (CRP) levels were 1.7-fold higher in the control group than in the recurrence and treatment failure group (BL p=0.018)."

10. "Results: What does LC and SC stand for?"
Please define these terms on first use.

RESPONSE:

Revised sentences on page 5, Methods and Material/Bacteriological assessment section:

"Moreover, sputum samples were cultured on solid Lowenstein-Jensen (LJ) media (solid culture, SC) and in liquid media using the BACTEC MGIT 960 system (liquid culture, LC)."

Reviewer #2 (Remarks to the Author):

The identification of reliable biomarkers for tuberculosis represents a critical advancement in the global effort to fight TB. Currently, microbiological tests such as culture remain the gold standard for monitoring treatment response; however, they are time-consuming, resource-intensive, and often fail to capture the complete clinical picture. Biomarkers offer the potential for a faster, more sensitive, and patient-centered approach to assess treatment efficacy and predict relapse risk. This is particularly important given the limitations of current tools in detecting viable bacilli and distinguishing between true cure and persistent subclinical disease. The development and validation of such biomarkers would not only enhance individualized patient care but also strengthen public health strategies aimed at reducing TB recurrence and transmission. This study proposes non-sputum biomarkers for predicting TB recurrence after treatment completion. Nevertheless, future validation in a larger cohort is required to confirm these findings. Importantly, the study provides a valuable foundation for future research aimed at improving post-treatment monitoring and long-term outcomes in TB patients.

RESPONSE to Reviewer 2:

We thank Reviewer 2 for the valuable and constructive suggestions that have helped improve our manuscript. In particular, we greatly appreciate the idea of including the correlation analysis, which we have now implemented and found to be very informative.

Major Concerns:

1. Clarification of MTB125

Please provide a clear explanation of what MTB125 refers to. Which specific peptides does it include?

RESPONSE: Thank you for your suggestion. The peptide pool consists of 125 non-redundant CD4⁺ T cell epitopes derived from M. tuberculosis antigens. These epitopes were identified

based on recognition by at least one donor in a South African cohort, following screening of 253 peptides previously shown to be immunogenic. Redundant and overlapping peptides mapping to the same epitope region were consolidated, resulting in a final set that captures the overall breadth of T cell responses.

We have clarified the definition of MTB125 in **the Methods section/ selection of PBMC samples and intracellular cytokine staining (page 5, paragraph1)** as follows:

“Of note, MTB125 is a peptide pool consisting of 125 selected immunogenic MTB peptides described in more detail by Lindestam Arlehamn et al.”

2. Inconsistency in Methods and Results

The Methods section mentions stimulation with ESAT6/CFP10, yet no corresponding data or results are presented in the manuscript. Could you clarify whether this analysis was performed and, if so, why the results are not included?

RESPONSE:

Thank you for pointing this out. In our experimental setup, we initially planned to stimulate PBMCs with PPD, MTB125, SEB, ESAT6/CFP10, and CMV, in that order. However, due to limited cell numbers in many samples, it was not feasible to include all stimulations for a large portion of the cohort. ESAT6/CFP10 and CMV stimulations were often omitted, resulting in insufficient data for meaningful analysis. Given that we already faced limited sample sizes for PPD and MTB125, we ultimately decided not to pursue analysis of the ESAT6/CFP10 and CMV data.

To avoid confusion, we have now removed the reference to ESAT6/CFP10 and CMV from the Methods section, as no corresponding results are presented.

3. Statistical Analysis

Have you considered applying a univariate logistic regression to assess the association between individual variables and the binary outcome (e.g., relapse/no relapse)? Furthermore, a multivariate logistic regression could be valuable to evaluate the simultaneous effect of multiple independent variables on the outcome.

RESPONSE: We thank the reviewer for this thoughtful question to perform univariate and multivariate logistic regression analyses to assess associations between individual variables and the binary outcome. Our initial intent with Table 1 was primarily descriptive; to provide a clear overview of the case and control cohorts, including potential baseline imbalances that may inform the interpretation of our downstream analyses. We did not aim to infer predictive or causal relationships between clinical variables and the outcome. Furthermore, while the design of our study is well-suited for multivariate regression analysis, we believe that our current sample size is not adequately powered to support robust multivariate modeling without the risk of overfitting or generating unstable estimates. The frequency of recurrence after drug-susceptible TB treatment is low, and consequently the number of available recurrence cases in our study is very limited for such an approach. Specifically, there are 40 or fewer recurrence patients and matched controls respectively, depending on the number of clinical and numerical variables considered for the regression model. Recurrence patients exhibit significant diversity, and the timing of recurrence is likely to play an important role. This is particularly relevant when considering differences among non-converting patients, those who revert at the end of treatment, and those who experience a recurrence at a later time point. Splitting the data into these three categories would reduce the number of data points to less than 20 relapse cases per group. When paired with the variables that might be relevant for predicting relapse (e.g. sex, clinical site, severity of the disease, HIV infection and ART treatment, culture status, BMI,

and comorbidities) we believe that the available data will not be sufficient to draw meaningful conclusions.

Moreover, other studies and meta-analyses we referred to in the introduction have already explored the question of recurrence prediction using available clinical covariates with more extensive datasets and advanced statistical modeling approaches (<https://pubmed.ncbi.nlm.nih.gov/30397355/> , <https://thorax.bmj.com/content/74/3/291.short>; <https://doi.org/10.1164/rccm.200407-905OC>; <https://pmc.ncbi.nlm.nih.gov/articles/PMC10941165/>; <https://pubmed.ncbi.nlm.nih.gov/31881023/>).

The main aim of our study was to assess the performance of the selected biomarkers in relation to different unsuccessful TB treatment outcomes, rather than to identify clinical predictors of these outcomes. Given this objective, we respectfully prefer to limit our analyses to descriptive comparisons that support the interpretation of our biomarker findings.

4. Correlation Between Assays

Did you consider assessing the correlation between the TAM-TB assay results and RNA signatures across the different study groups? Such an analysis could strengthen the interpretation of the findings.

RESPONSE:

We appreciate this insightful suggestion. To address it, we conducted a correlation analysis between TAM-TB assay results and RNA signatures, which has been incorporated into the Results section (**page 10, Results/ Level of MTB-specific CD4 T-cell activation correlates with transcriptomic signature scores across patient groups**) and is presented as **Figure 6**.

Further, we added a statement on a potential combined biomarker approach to the discussion (**page 12, paragraph 3**).

“Correlation analyses between transcriptomic signatures and TAM-TB showed moderate agreement, suggesting that combining these biomarkers may have the potential to improve diagnostic precision for TB outcomes.”

Response to Reviewers

Manuscript Title: Host response biomarkers of tuberculosis recurrence and treatment failure

Manuscript ID: COMMSMED-25-1036B

Journal: Communications Medicine

Date: 30 November 2025

Dear Editors and Reviewers,

We are grateful for the reviewers' positive assessments and their constructive input, which have helped us strengthen the manuscript. We have implemented the final minor changes as requested. All changes are highlighted in yellow for your convenience. In addition, we have uploaded a clean version of the manuscript without highlighting it. Our point-by-point responses are detailed below.

We have uploaded the completed Revision Instructions and, as requested, added a Plain Language Summary, an Editorial Summary, and the Communication in Medicine and STAT checklists to the manuscript. We thank you again for taking the time to review our revised manuscript.

Bernadette Bauer, corresponding author, on behalf of all authors

Reviewer #1:

Congratulations on an excellent revision and a thorough, thoughtful rebuttal. Your responses comprehensively address my prior concerns and considerably strengthen the manuscript. I have only two minor follow-ups there's a small typo in the Statistical Methods—please change “Walt” to “Wald”. For data availability, please deposit the RNA-seq data in a public, field-appropriate repository and update the Data Availability statement with the accession IDs; likewise, please provide a public link/DOI for the analysis code.

RESPONSE to Reviewer 1: We sincerely thank the reviewer for the positive feedback and kind words. We appreciate the recognition of our revisions and thoughtful responses. We have now addressed the two minor follow-ups as requested, as detailed below.

1. there's a small typo in the Statistical Methods—please change “Walt” to “Wald”;

RESPONSE: Thanks for pointing this out we corrected the typo in the Statistical analysis section on page 13.

2. For data availability, please deposit the RNA-seq data in a public, field-appropriate repository and update the Data Availability statement with the accession IDs; likewise, please provide a public link/DOI for the analysis code.

RESPONSE: We have updated both the Data Availability and Code Availability statements. The data are now publicly accessible, and the location where they can be accessed is clearly indicated in the revised manuscript page 14.

Reviewer #2 (Remarks to the Author):

The authors have satisfactorily addressed the reviewers' comments and provided thorough justifications in cases where the requested analyses could not be performed.

RESPONSE to Reviewer 1::

We sincerely thank the reviewer for the positive evaluation and for acknowledging our responses and justifications. We appreciate the constructive feedback provided throughout the review process, which has helped us improve the quality and clarity of the manuscript.